# Tunable molecular tension sensors reveal extension-based control of vinculin loading

**Andrew S LaCroix, Andrew D Lynch, Matthew E Berginski, Brenton D Hoffman\***

Department of Biomedical Engineering, Duke University, Durham, United States

**Abstract** Molecular tension sensors have contributed to a growing understanding of mechanobiology. However, the limited dynamic range and inability to specify the mechanical sensitivity of these sensors has hindered their widespread use in diverse contexts. Here, we systematically examine the components of tension sensors that can be altered to improve their functionality. Guided by the development of a first principles model describing the mechanical behavior of these sensors, we create a collection of sensors that exhibit predictable sensitivities and significantly improved performance *in cellulo*. Utilized in the context of vinculin mechanobiology, a trio of these new biosensors with distinct force- and extension-sensitivities reveal that an extension-based control paradigm regulates vinculin loading in a variety of mechanical contexts. To enable the rational design of molecular tension sensors appropriate for diverse applications, we predict the mechanical behavior, in terms of force and extension, of additional 1020 distinct designs.

DOI: https://doi.org/10.7554/eLife.33927.001

**\*For correspondence:**
brenton.hoffman@duke.edu

**Competing interests:** The authors declare that no competing interests exist.

## Introduction

The ability of cells to generate and respond to mechanical loads is increasingly recognized as a critical driver in many fundamentally important biological processes, including migration (*Doyle et al., 2009*; *Lo et al., 2000*; *Pelham and Wang, 1997*), proliferation (*Chen et al., 1997*; *Provenzano and Keely, 2011*), differentiation (*Engler et al., 2006*; *Heo et al., 2016*; *McBeath et al., 2004*), and morphogenesis (*Heisenberg and Bellaïche, 2013*; *Wozniak and Chen, 2009*). While the mechanosensitive signaling pathways enabling these responses are poorly understood, most are thought to have a common basis: the mechanical deformation of load-bearing proteins (*Cost et al., 2015*; *Hoffman et al., 2011*; *Ju et al., 2016*). As such, several technologies for measuring the loads borne by specific proteins in living cells have emerged (*Freikamp et al., 2016*; *Hoffman, 2014*; *LaCroix et al., 2015b*; *Liu et al., 2017*; *Polacheck and Chen, 2016*). These biosensors, collectively referred to as molecular tension sensors, leverage the distance-dependence of Förster Resonance Energy Transfer (FRET) to measure the extension of and, if properly calibrated, the forces across a specific protein of interest (*Austen et al., 2013*; *Freikamp et al., 2017*; *Hoffman, 2014*; *LaCroix et al., 2015a*). For example, using this approach, the tension across vinculin was shown to regulate a mechanosensitive switch governing the assembly/disassembly dynamics of focal adhesions (FAs) (*Grashoff et al., 2010*). While this and several other FRET-based molecular tension sensors provide a critical view into mechanosensitive processes (*Cost et al., 2015*; *Jurchenko and Salaita, 2015*), fundamental questions regarding the nature and the degree of the mechanical loading of proteins remain. A key limitation has been the inability to create tension sensors with diverse mechanical sensitivities suitable for a wide variety of biological applications (*Freikamp et al., 2017*).

To date, genetically-encoded molecular tension sensor modules (TSMods), which are incorporated into various proteins to form distinct tension sensors (*Figure 1A*), have been created without *a*

**eLife digest** Cells must sense signals from their surroundings to play their roles within the body. These signals can be biochemical, such as growth-promoting substances, or mechanical, for example the stiffness or softness of the environment.

Mechanical signals can be detected by load-bearing proteins, which stretch like tiny springs in response to forces. In animals, these proteins span the membrane separating the interior of the cell from the exterior. Externally, the proteins attach to structures around the cell; internally, they connect to the machinery that both generates forces and allows cells to respond to signals from outside. As such, load-bearing proteins form a direct mechanical link between cell and environment.

Scientists use tools called molecular tension sensors to measure how much a load-bearing protein stretches in response to changes, and the force that is being applied to it. However, just like any other type of scale, these sensors only work over a certain range, which happens to be limited. This means that, for example, they cannot measure forces in tissues that are too soft (like the brain), or too stiff (such as bones). New sensors that can assess forces in these contexts are therefore needed, but so far research in this area has been slow due to a reliance on 'trial-and-error' approaches.

Here, LaCroix et al. developed a new method to predict the sensitivity of molecular tension sensors inside cells. This was accomplished by examining several existing sensors, and identifying which components could be altered to change the properties of the sensors. Then, this information was used to create a computer model that could predict how new sensors would behave, and which range of forces they could measure. Finally, the sensors designed following this method were tested in mouse cells grown in the laboratory, and they worked better than their predecessors.

The next step was for LaCroix et al. to use a trio of new sensors with different sensitivities to study the load-bearing protein vinculin in mouse cells. The goal was to figure out exactly how cells manage their load-bearing proteins. Indeed, it was widely assumed that a cell acts on a load-bearing protein by applying a force on it. In response, the protein would stretch by a certain amount, which can change depending on its properties – a 'stiffer' protein would stretch less. Unexpectedly, the new sensors showed that cells instead manipulate how much vinculin stretches, applying varying forces to achieve the same length of the protein in different environments.

Improved molecular tension sensors will give scientists a better insight into how cells respond to their mechanical environment, which could help to direct cell behavior in tissues engineered in the laboratory. This knowledge is also directly relevant to human health, as the mechanical properties of many tissues change during disease, such as tumors stiffening during cancer.

DOI: https://doi.org/10.7554/eLife.33927.002

*priori* knowledge of their mechanical sensitivity. TSMod development has largely relied on a biologically-inspired 'guess-and-check' design approach using naturally-occurring extensible polypeptides or protein domains as deformable elements in the FRET-based tension sensors. Furthermore, despite the use of these sensors to study intracellular processes, calibration measurements of their mechanical sensitivity are typically performed in vitro using highly precise single molecule techniques. Reported force sensitivities of several in vitro calibrated TSMods are 1–6 pN (*Grashoff et al., 2010*), 2–11 pN (*Brenner et al., 2016*), 3–5 pN (*Ringer et al., 2017*), 6–8 pN (*Austen et al., 2015*), or 9–11 pN (*Austen et al., 2015*). However, it is unclear if these ranges are sufficient for diverse mechanobiological studies, and the applicability of these in vitro calibrations to sensors that are utilized *in cellulo* has not been verified.

We sought to overcome these limitations by creating new TSMods that do not rely on naturally occurring extensible domains or in vitro calibration schemes. These new TSMods consist of a Clover-mRuby2 FRET pair connected by unstructured polypeptide extensible domains of various lengths. As the entropy-driven mechanical resistance of unstructured polypeptides can be accurately predicted by established models of polymer extension (*Becker et al., 2010*), the force- and extension-sensitivities can be determined independently of in vitro calibration experiments. Using these advancements, we generate a variety of new tension sensors for the FA protein vinculin. These include a version optimized for sensitivity, which shows a nearly 3-fold increase in performance, as well as a suite of sensors with distinct mechanical sensitivities capable of determining if vinculin

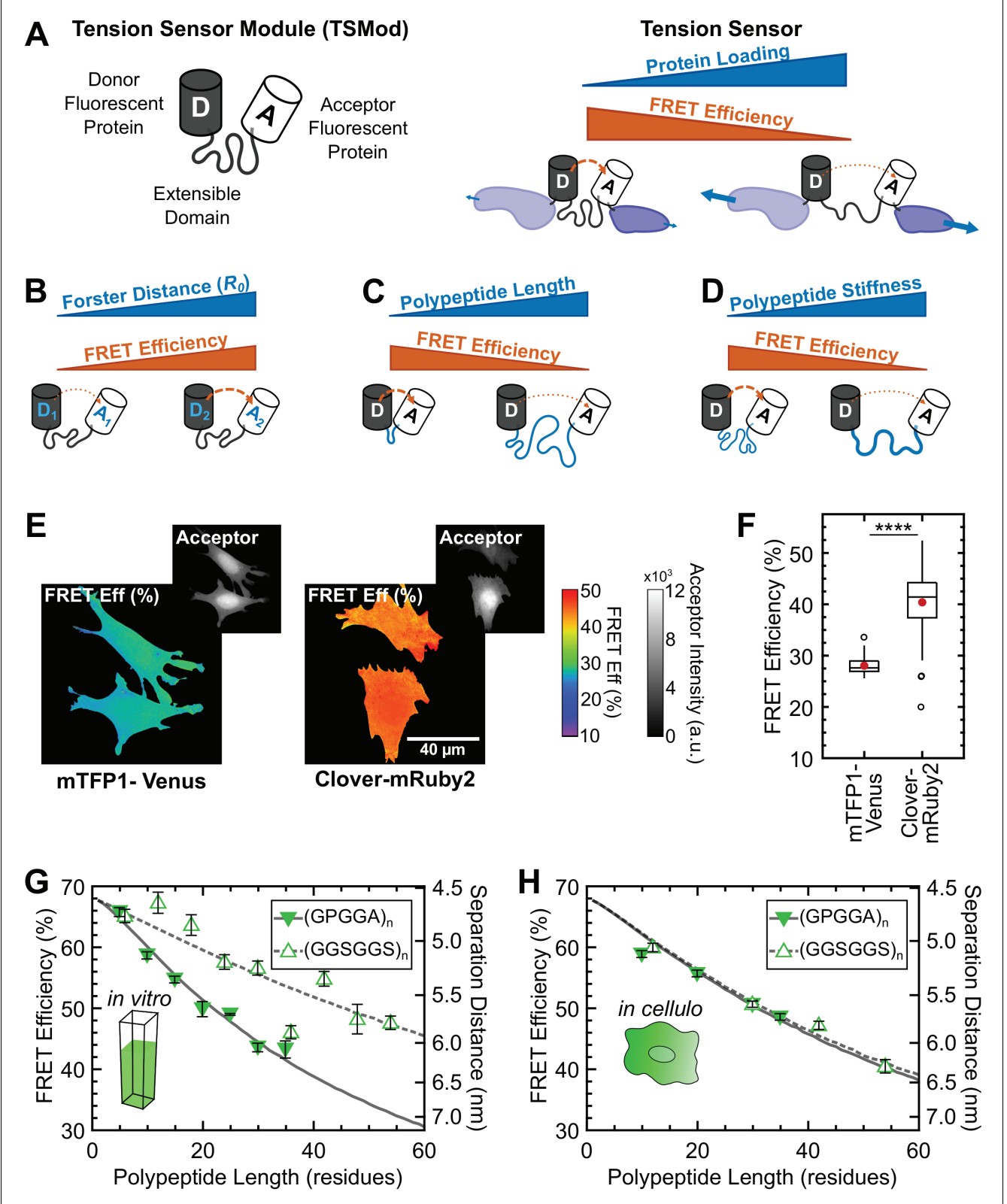

**Figure 1.** Design and characterization of tunable FRET-based molecular tension sensors. (**A**) Schematic depiction of a generic TSMod and inverse relationship between FRET and force for molecular tension sensors under tensile loading. (**B-D**) TSMod function depends on the Förster radius of the chosen FRET pair (**B**) as well as the length (**C**) and stiffness (**D**) of the extensible polypeptide domain. (**E**) Representative images of soluble mTFP1-Venus and Clover-mRuby2 TSMods expressed in Vin-/- MEFs. (**F**) Quantification of unloaded FRET efficiency for mTFP1-Venus and Clover-mRuby2

*Figure 1 continued on next page*

*Figure 1 continued*

TSMods; (n = 53 and 92 cells, respectively); red filled circle denotes sample mean; **** p < 0.0001, Student's t-test, two-tailed, assuming unequal variances. (**G**) Quantification of FRET-polypeptide length relationship for minimal Clover-mRuby2 TSMods in vitro; each point represents data from at least 5 independent experiments; lines represent model fits where $L_P$ is the only unconstrained parameter. (**H**) Quantification of FRET-polypeptide length relationship for minimal Clover-mRuby2 based TSMods *in cellulo*; each point represents at least n = 48 cells from three independent experiments; lines represent model fits where $L_P$ is the only unconstrained parameter. All error bars, s.e.m.

DOI: https://doi.org/10.7554/eLife.33927.003

The following source data and figure supplements are available for figure 1:

**Source data 1.** Measurements and models of the mechanical behavior of TSMods in vitro and in cellulo.

DOI: https://doi.org/10.7554/eLife.33927.010

**Figure supplement 1.** FRET efficiency measurements depend on the presence of unstructured residues in FPs, but are insensitive to fixation and sensor intensity.

DOI: https://doi.org/10.7554/eLife.33927.004

**Figure supplement 1—source data 1.** FRET-length relationships for TSMods in various conditions.

DOI: https://doi.org/10.7554/eLife.33927.005

**Figure supplement 2.** Increase in unloaded FRET efficiency with Clover-mRuby2 sensors in vitro.

DOI: https://doi.org/10.7554/eLife.33927.006

**Figure supplement 2—source data 1.** Fluorometric FRET measurements.

DOI: https://doi.org/10.7554/eLife.33927.007

**Figure supplement 3.** 'Minimal' FPs exhibit spectral properties indistinguishable from full-length parent FPs.

DOI: https://doi.org/10.7554/eLife.33927.008

**Figure supplement 3—source data 1.** Fluorescent protein spectra.

DOI: https://doi.org/10.7554/eLife.33927.009

---

loading is subject to extension-based or force-based control. Lastly, we computationally predict the mechanical behavior expected for a variety of unstructured polypeptide-based tension sensors for several common FRET pairs. This resource should allow for the expedited creation and rational design of molecular tension sensors suited for use in diverse contexts, alleviating a significant limitation in the study of mechanobiology.

## Results

### Creation of TSMods based on synthetic unstructured polypeptides

TSMods for intracellular use consist of two fluorescent proteins (FPs) connected by an extensible domain (*Figure 1A*). To enable the creation of tension sensors with diverse mechanical sensitivities, we constructed a variety of TSMods using FPs with distinct photophysical properties connected by unstructured polypeptides of various lengths and mechanical properties, as each of these character-istics critically determine the behavior of these sensors (*Figure 1B–D*). We based our designs on the first calibrated TSMod (*Grashoff et al., 2010*), which is comprised of the mTFP1-Venus FRET pair connected by a flagelliform silk-based polypeptide with the repeated sequence (GPGGA)$_8$, and has been used in a variety of tension sensors (*Cost et al., 2015*; *Jurchenko and Salaita, 2015*).

First, we evaluated the role of the FPs in TSMod function. Reasoning that increases in the unloaded FRET efficiency could potentially increase the dynamic range of the sensor as well as allevi-ate technical issues with measuring small FRET signals, we sought to increase the FRET efficiency in this state (*Figure 1B*). To do so, we replaced mTFP1-Venus with the green-red FRET pair Clover-mRuby2 (*Lam et al., 2012*), which exhibits stronger FRET at a given separation distance (Förster radius ($R_0$) of 5.7 and 6.3 nm, respectively). This simple substitution yielded a 12% higher baseline (unloaded) FRET efficiency that was observed in fixed (*Figure 1E, F*) and live cells (*Figure 1—figure supplement 1*), as well as cell lysates (*Figure 1—figure supplement 2*). While the benefits of the improved photophysical properties of Clover-mRuby2 are established (*Lam et al., 2012*), we probed the effect of the physical structure of the FPs on their performance in TSMods. Although commonly identified by a characteristic beta-barrel structure, FPs also contain short unstructured regions at their termini that likely contribute to the effective mechanical properties of the extensible domains used in TSMods (*Figure 1—figure supplement 1*) (*Ohashi et al., 2007*). Previous work, and our

data (*Figure 1—figure supplement 3*), have shown that 'minimal' Clover (residues 1 – 227) and mRuby2 (residues 3 – 236) exhibit absorbance and emission spectra indistinguishable from their full-length counterparts (*Austen et al., 2015*; *Li et al., 1997*; *Ohashi et al., 2007*; *Ouyang et al., 2008*; *Shimozono et al., 2006*). Therefore, to mitigate concerns about FPs affecting the mechanical properties of the extensible domains and further increase the unloaded FRET efficiency, minimal versions of Clover and mRuby2 were used in the construction of all TSMods.

Recent evidence suggests that both the mechanical properties (*Austen et al., 2013*; *Ringer et al., 2017*) and the length (*Brenner et al., 2016*) of the extensible domain provide viable means by which to tune the mechanical sensitivity of TSMods (*Figure 1C,D*). Towards this end, we created a variety of TSMods containing extensible domains comprised of either the flagelliform-based (GPGGA)$_n$, which is thought to be relatively stiff (*Becker et al., 2003*), or the synthetic (GGSGGS)$_n$ which has been characterized as an unstructured polypeptide and has previously been employed as a tunable linker in biochemical sensors (*Evers et al., 2006*). Analysis of TSMods in cell lysates showed that those with (GGSGGS)$_n$ extensible domains exhibit higher FRET efficiencies than those with (GPGGA)$_n$ extensible domains of the same length (*Figure 1G*), suggesting that (GPGGA)$_n$-based polypeptides are indeed stiffer, and thus force the FPs apart more readily, than (GGSGGS)$_n$-based polypeptides. However, when (GPGGA)$_n$ and (GGSGGS)$_n$ TSMods were evaluated *in cellulo*, the FRET efficiency versus length relationships were indistinguishable, suggesting the polypeptides are exhibiting identical mechanical properties (*Figure 1H*). Together, these data demonstrate that factors dictating sensor functionality in the absence of applied load can be environmentally sensitive, and that the behavior of TSMods observed in vitro may not reflect their behavior *in cellulo*. As such, these results raise concerns about the applicability of calibrations of FRET-based tension sensors performed in vitro to sensors that are used in intracellular environments (further discussed in Appendix 1).

## A quantitative model describing the mechanical sensitivity of TSMods

As an alternative to TSMod calibration through in vitro approaches, we pursued a modeling-based approach for describing the mechanical sensitivities of TSMods. Given that FPs linked by (GGSGGS)$_n$ polypeptides are well-described by established models of polymer physics in unloaded conditions (*Evers et al., 2006*), we developed an analogous model to predict TSMod behavior under load. Briefly, the proposed calibration model incorporates three main aspects of TSMods: (1) the photophysical properties of the FRET pair (Förster radius, $R_0$), (2) the radii of the FPs ($R_{FP}$), and (3) the mechanical response of the extensible domain, which is well-described as a semi-flexible polymer by a persistence length ($L_P$) and a contour length ($L_C$) in the framework of the worm-like chain model (*Becker et al., 2010*). This modeling-based approach enables the prediction of the *in cellulo* mechanical response of FRET-based tension sensors by leveraging separate measurements of the *in cellulo* $L_P$ of the unstructured polypeptide used as the extensible domain. A detailed description of the development and implementation of the model, as well as comparison to other estimates of TSMod behavior are presented in Appendix 1, which refers to data presented in *Figure 2—figure supplement 1,2* and *Supplementary file 1*.

To validate this model, we first investigated its ability to describe the behavior of several types of TSMods in terms of the relationship between FRET and the length of the extensible domain in unloaded conditions. These measurements are critical in that they are used to estimate the mechanics of the extensible domain in terms of its persistence length $L_P$. To do so, estimates of $R_0$ and $R_{FP}$ were obtained from the literature, $L_C$ was directly calculated from the number of amino acids comprising the extensible domain, and $L_P$ was used as the single adjustable parameter. With only $L_P$ left unconstrained, the model accurately describes the behavior of TSMods containing (GPGGA)$_n$ and (GGSGGS)$_n$ extensible domains in unloaded conditions in *in vitro* (*Figure 1G*) and *in cellulo* (*Figure 1H*) environments with physically reasonable estimates of $L_P$. Model fits and 95% confidence intervals confirm that $L_P$ estimates for (GPGGA)$_n$ and (GGSGGS)$_n$ polypeptides are significantly different *in vitro* ($0.74 \pm 0.05$ and $0.33 \pm 0.05$ nm, respectively), and collapse to one intermediate value *in cellulo* ($0.50 \pm 0.02$ and $0.48 \pm 0.05$ nm, respectively). Also, to demonstrate that the literature estimates of $R_0$ and $R_{FP}$ were appropriate, we performed a sensitivity analysis, leaving either $R_{FP}$ or $R_0$ unconstrained. We observe negligible improvement in fit quality and achieve similar estimates of $L_P$ (*Figure 2—figure supplement 3* and *4*), validating our approach. Overall, these results demonstrate the functionality of the model to measure the $L_P$ of TSMod extensible domains in unloaded

conditions and also suggest that the observed mechanics of the extensible domain can change in different environments, although less-so for (GGSGGS)$_n$ polypeptides.

Next, we sought to investigate the generalizability of the model as well as validate the ability of the model to describe the behavior of TSMods subject to tensile loads. Therefore, we examined model fits to published fluorescence-force spectroscopy measurements of Cy3-Cy5 dyes linked by (GPGGA)$_n$ extensible domains (*Brenner et al., 2016*). Again, with only $L_P$ unconstrained, the model accurately describes the behavior of these TSMod-like constructs in both unloaded conditions (*Figure 2A*) and under tensile loads (*Figure 2B*). Importantly, each of these datasets is well-described by the same persistence length ($L_P = 1.05$ nm) indicating that the same mechanical model is appropriate for describing the behavior of unstructured polypeptides in both unloaded and loaded conditions when both measurements are determined in the same environment. For comparison, we show fits for a range of $L_P$ values from $1.0$ to $1.15$ nm (lines in *Figure 2A*, shaded region in *Figure 2B*). The robustness of these fits to various parameter constraints was also verified (*Figure 2—figure supplement 5*). It is important to note that these differences in the $L_P$ of (GPGGA)$_n$

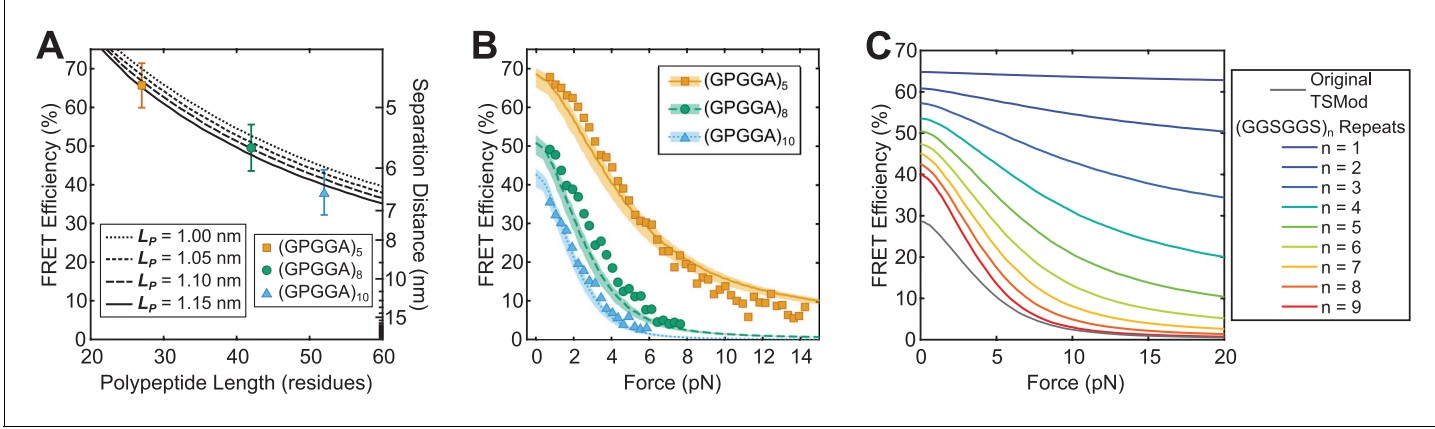

**Figure 2.** Predicting TSMod calibrations using a biophysical model. (A, B) Model descriptions, at various persistence lengths ($L_P$), of FRET-polypeptide length relationship (A) and FRET-force responses (B) of Cy3 and Cy5 dyes linked by SMCC linker + cysteine modified (GPGGA)$_n$ polypeptides; data was digitized based on histograms from (*Brenner et al., 2016*); model parameters $R_0 = 5.4$ nm, $R_{FP} = 0.95$ nm, $L_P = 1.05$ nm (range 1.00 to 1.15 nm); error bars, s.d. (C) Model predictions of force sensitivity of TSMods comprised of Clover-mRuby2 FRET pair and (GGSGGS)$_n$ extensible domains in comparison to the original TSMod (*Grashoff et al., 2010*); model parameters $R_0 = 6.3$ nm, $R_{FP} = 2.3$ nm, $L_P = 0.48$ nm.
DOI: https://doi.org/10.7554/eLife.33927.011

The following source data and figure supplements are available for figure 2:

**Source data 1.** Measurements and models of the mechanical behavior of TSMod-like constructs in vitro.
DOI: https://doi.org/10.7554/eLife.33927.020

**Figure supplement 1.** Schematic depiction of biophysical model describing the mechanical sensitivity of TSMods.
DOI: https://doi.org/10.7554/eLife.33927.012

**Figure supplement 2.** Verification of the proper implementation of a biophysical model describing the mechanical sensitivity of FRET-based TSMods.
DOI: https://doi.org/10.7554/eLife.33927.013

**Figure supplement 2—source data 1.** Numerical simulations of the mechanical behavior of worm-like chains.
DOI: https://doi.org/10.7554/eLife.33927.014

**Figure supplement 3.** Parameter constraint has minimal effects on measurement of polypeptide persistence length ($L_P$) in vitro.
DOI: https://doi.org/10.7554/eLife.33927.015

**Figure supplement 4.** Parameter constraint has minimal effects on measurement of polypeptide persistence length ($L_P$) *in cellulo*.
DOI: https://doi.org/10.7554/eLife.33927.016

**Figure supplement 5.** Parameter constraint has minimal effects on measurement of polypeptide persistence length ($L_P$) for TSMod-like constructs in unloaded or loaded conditions.
DOI: https://doi.org/10.7554/eLife.33927.017

**Figure supplement 6.** Experimental and theoretical examinations of other models (*Brenner et al., 2016*) of (GPGGA)$_n$ mechanical sensitivity.
DOI: https://doi.org/10.7554/eLife.33927.018

**Figure supplement 6—source data 1.** Comparitive analysis of models of flagelliform polypeptide mechanics.
DOI: https://doi.org/10.7554/eLife.33927.019

polypeptides in various environments (*Figure 2G,H*, *Figure 2A*) support the idea that in vitro calibrations should be applied to sensors used in living cells, or in different in vitro environments, with caution.

## A novel approach for predicting the *in cellulo* calibration of TSMods

Together these results suggest a simple model-based calibration scheme by which measurements of extensible domain mechanics ($L_P$) in unloaded conditions are utilized to predict TSMod behavior under tensile loading. While our modeling efforts indicate that both $(GPGGA)_n$ and $(GGSGGS)_n$ polypeptide mechanics are consistent with unstructured polypeptides (*Figure 2—figure supplement 6*), we only generate calibration predictions for TSMods containing $(GGSGGS)_n$ extensible domains because they are also less sensitive to environmental changes. In the context of the model, the *in cellulo* persistence length of the $(GGSGGS)_n$ extensible domain ($L_P = 0.48$ nm, *Figure 1H*) is combined with literature estimates of the radii (*Hink et al., 2000*) and photophysical properties (*Lam et al., 2012*) of Clover and mRuby2 to predict the response of $(GGSGGS)_n$-based TSMods under applied loads (*Figure 2C*). This model-based calibration scheme uniquely overcomes the environmental sensitivity of the extensible domain (compare *Figure 1G and H*) by allowing for *in cellulo* measurements of $L_P$ to be used to estimate the mechanical sensitivity of TSMods.

## Optimized tension sensor reveals gradients of vinculin tension across FAs

To determine which extensible domain length will be optimal for measuring tension across vinculin, we evaluated TSMod mechanical sensitivity across different force regimes by calculating the derivative along the FRET-force curve (*Freikamp et al., 2017*) (*Figure 2C*, *Figure 3—figure supplement 1*). Given the original vinculin tension sensor (VinTS) reported average loads of ~2.5 pN across vinculin that varied from 1 to 6 pN (*Grashoff et al., 2010*), we choose to further investigate the performance of the TSMod containing the nine-repeat extensible domain, as it exhibits the highest sensitivity in this force regime and is capable of capturing the distribution of the loads on vinculin (*Figure 3—figure supplement 1A*). This nine-repeat linker also provides a good balance between FRET dynamic range and peak sensitivity (*Figure 3—figure supplement 1B,C*). An optimized VinTS (opt-VinTS) was created by genetically inserting this TSMod into vinculin at same site, after amino acid 883, as in the original VinTS design (*Grashoff et al., 2010*).

We assessed the performance of opt-VinTS by evaluating its ability to detect changes in vinculin loading across both subcellular and FA length scales. Vin-/- MEFs expressing either VinTS or opt-VinTS showed indistinguishable cell and FA morphologies (*Figure 3A,A', C,C'*, *Figure 3—figure supplement 2*). Furthermore, line scans of acceptor intensity across single FAs indicated similar localization of each sensor (*Figure 4—figure supplement 3A'', A''', C'', C'''*). These findings indicate that the two sensors exhibit identical biologically functionality. At a subcellular length scale, consistent with our previous findings (*Rothenberg et al., 2015*), both VinTS and opt-VinTS report highest loads (lowest FRET efficiency) in the cell periphery, and no appreciable tensile loading of vinculin in the cell center (*Figure 3B,D*). Based on previous reports of gradients of vinculin loading within individual FAs (*Sarangi et al., 2017*) and a skewed distribution of mechanical stresses at the cell-substrate interface (*Blakely et al., 2014*; *Legant et al., 2013*; *Morimatsu et al., 2015*; *Plotnikov et al., 2012*), we expected to see similar distally-skewed vinculin tensions. Such gradients are difficult to discern in peripheral FAs of cells expressing the original tension sensor (*Figure 3B', B'', B'''*). However, striking gradients of vinculin tension across single FAs were clearly visible in peripheral FAs of cells expressing opt-VinTS (*Figure 3D', D'', D'''*). To quantitatively gauge performance, we quantified the FRET efficiency change across length-normalized FAs (slope, *Figure 3B''', D'''*). This analysis revealed an almost 3-fold improvement in the performance opt-VinTS (slope = 15.0%/FA) when compare the original design (slope = 5.5%/FA). In total, these results show that, as predicted by the model, opt-VinTS is significantly more sensitive than the original VinTS.

## Vinculin loading is subject to an extension-based control mechanism

A central premise of mechanotransduction, the process by which cells sense and respond to mechanical stimuli, is that applied loads induce conformational changes in mechanosensitive proteins, leading to biochemically distinct functions. However, it is unknown whether the forces or the

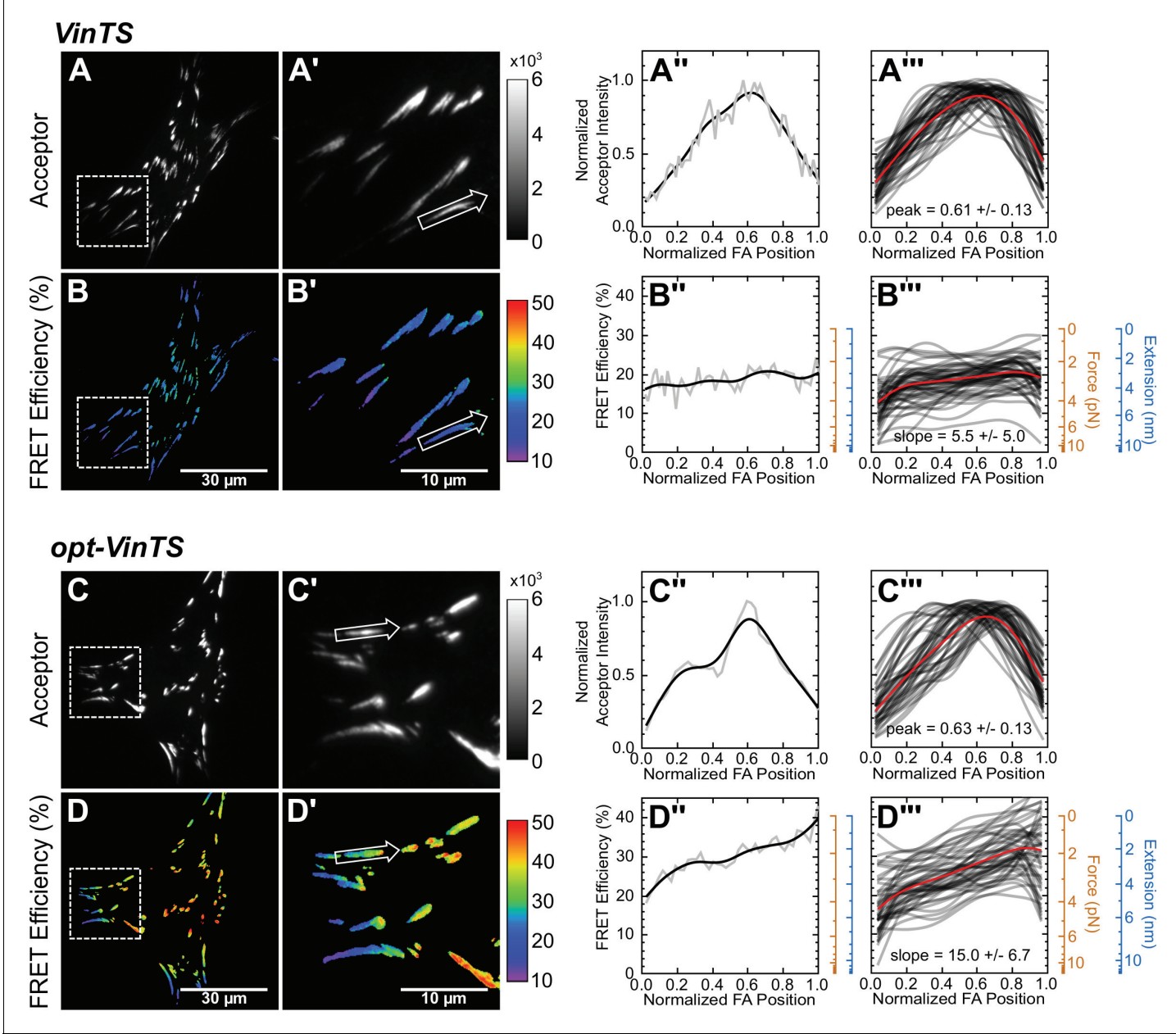

**Figure 3.** Optimized tension sensor reveals sub-FA gradients in vinculin tension. (panels **A** and **B**) Representative images of subcellular distribution of VinTS (*Grashoff et al., 2010*) (**A**, **A'**) along with representative (**A''**) and aggregate (**A'''**) line scans of single FAs of size >0.5 µm$^2$ and axis ratio >1.5; corresponding masked FRET images (**B**, **B'**) and line scans (**B''**, **B'''**); n = 55 FAs from n = 11 cells from three independent experiments; averaged intensity and FRET profiles in red. (panels **C** and **D**) As in panels (**A** and **B**), but with optimized opt-VinTS construct; n = 49 FAs from n = 10 cells from three independent experiments.

DOI: https://doi.org/10.7554/eLife.33927.021

The following figure supplements are available for figure 3:

**Figure supplement 1.** Selection of optimal (GGSGGS)$_n$ extensible domain length in a Clover-mRuby2 based TSMod for measuring ~1-6 pN loads borne by vinculin.

DOI: https://doi.org/10.7554/eLife.33927.022

**Figure supplement 2.** FA morphologies, cell morphologies, and sensor localization to FAs are indistinguishable between different versions of VinTS.

DOI: https://doi.org/10.7554/eLife.33927.023

extensions experienced by proteins mediate the activation of mechanosensitive signaling pathways. Experimental evaluation of this molecular-scale question has been challenging because force and extension are inherently linked. For example, in the case of molecular tension sensors, the force-extension relationship for the extensible domain is monotonic, so any given force corresponds to a unique extension (*Figure 4—figure supplement 1*). Note that extension refers to the change in the average length ($\langle r_z \rangle$, Appendix 1, *Equation 6*), not the separation distance of the FPs ($r_c$) in the construct. Importantly, $\langle r_z \rangle$ is independent of the size of the extensible domain.

To determine whether conserved forces or extensions mediate vinculin loading, we created a trio of vinculin tension sensors with extensible domains comprised of five, seven, or nine repeats of (GGSGGS)$_n$. As each sensor has a unique force-extension curve, the application of equivalent forces to the three constructs will result in three distinct extensions, and vice versa (*Figure 4—figure supplement 1*). Cells expressing equivalent amounts of each sensor spread and formed FAs normally (*Figure 4A–D*, *Figure 3—figure supplement 2*). Using the *in cellulo* calibration predictions described above (shown in *Figure 2C*), measured FRET efficiencies (*Figure 4E–H*) were converted to sensor forces (*Figure 4I–L*) and extensions (*Figure 4M–P*). Intriguingly, we observed similar distributions of extension (*Figure 4P*), and distinct distributions of tensile forces (*Figure 4L*) in FAs formed in cells expressing the various sensors. Furthermore, highly loaded FAs in the cell periphery exhibit conserved gradients in extension rather than force (*Figure 4—figure supplement 2*). To test whether vinculin is exclusively regulated by an extension-based paradigm, we conducted three additional control paradigm experiments. First, treatment of cells with the Y-27632 showed that vinculin extension-control is robust to short-term inhibition of ROCK-mediated cytoskeletal contractility (*Figure 4—figure supplements 3* and *4*). Secondly, vinculin extension-control does not require vinculin-talin interactions, as assessed through the introduction of a point mutation (A50I) in the three versions of VinTS (*Figure 4—figure supplements 5* and *6*). Finally, we observed that extension-based control still occurs on substrates of physiologically-relevant 10 kPa stiffness (*Figure 4—figure supplements 7* and *8*). Together, these results strongly suggest that loads across vinculin are exclusively governed by an extension-based control rather than the more commonly assumed force-based control paradigm.

To gain insight into the physical origins of force- versus extension-controlled loading of proteins within FAs, we examined how forces and extensions propagate through a simple structural model of a FA (see Appendix 2 for details and more comprehensive discussion of model results). Briefly, the structural model is comprised of various numbers of two distinct elements, which can be thought of as mechanically-dominant proteins or protein complexes within FAs. A sensor element (subscript 'S') and an alternative linker element (subscript 'L') are arranged in two layers (*Figure 4—figure supplement 9A*) meant to simulate the stratified organization of FAs (*Kanchanawong et al., 2010*). By comparing the relative variances in forces and extensions observed across sensor elements within various arrangements (*Figure 4—figure supplement 9B*), we examined whether a force-controlled or extension-controlled loading of the sensor element would be observed following a bulk force or extension input to the entire structure, and whether this depended on either the relative molecular abundance or the relative stiffness of each element (*Figure 4—figure supplement 9C*).

Regardless of the relative abundance of the elements or their respective stiffness, a force input to the entire structure always resulted in force-based control within the sensor elements. In contrast, an extension input to the entire structure, as might arise due to defined myosin motor step size (*Murphy et al., 2001*) or actin polymerization (*Peskin et al., 1993*), gave rise to both extension-controlled and force-controlled regimes in the sensor elements. The extension-controlled loading of the sensor element is more strongly observed when the sensor element is relatively soft and/or in low abundance, otherwise a force-controlled system is predicted (*Figure 4—figure supplement 9D*). Furthermore, in the extension-controlled regime, this simple model also predicts the linear relationship between sensor element stiffness and the force borne by the three sensor elements that was observed in all control paradigm experiments (*Figure 4—figure supplement 9E*). Together, these results demonstrate that protein extension, instead of applied force, might be a key mechanical variable in some mechanosensitive processes.

## Roadmap for future TSMod design

By expanding the simulated parameter space, the calibration model can also be used to predict the *in cellulo* mechanical sensitivity of a wide variety of potential TSMod designs. Specifically, as each

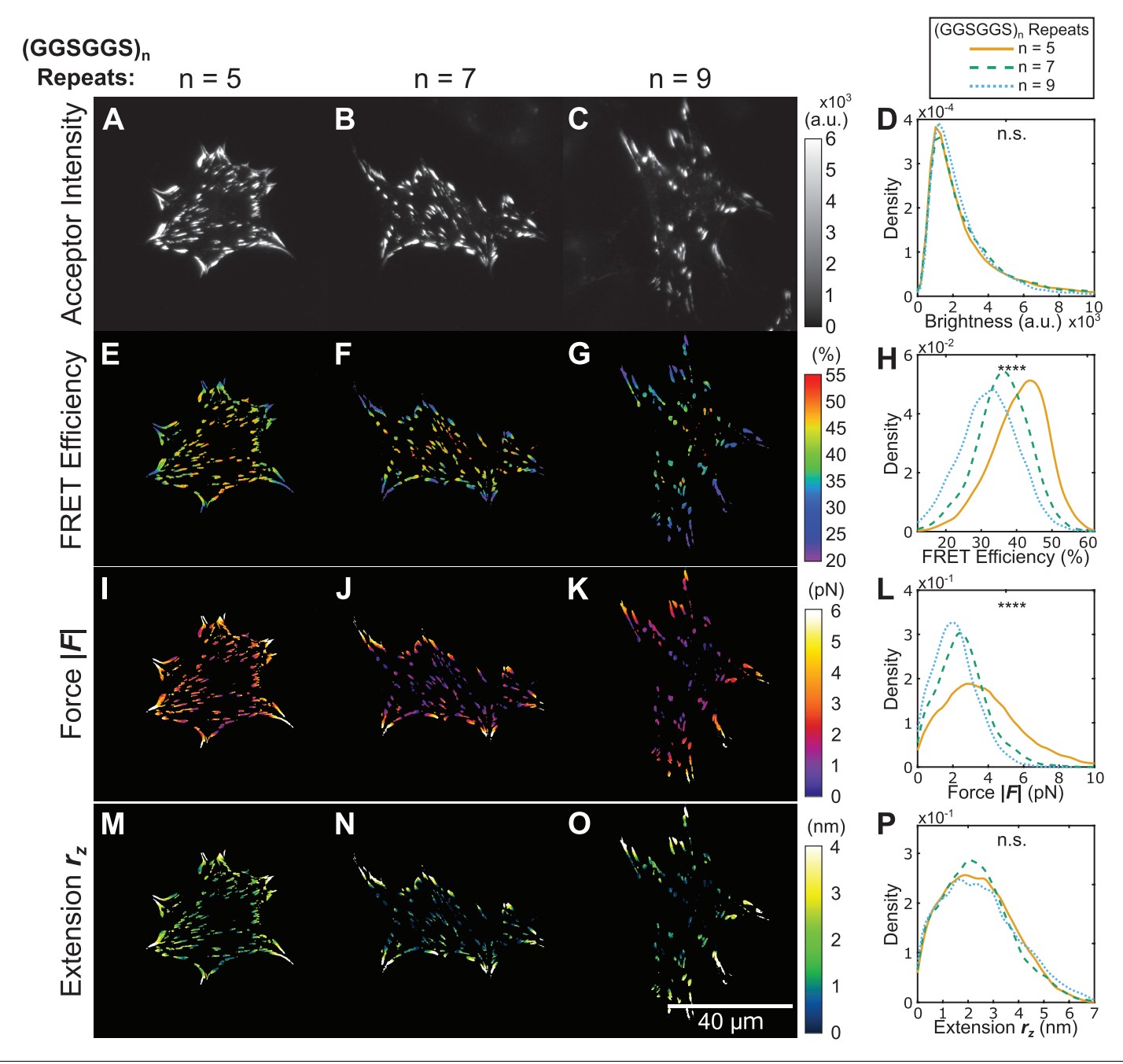

**Figure 4.** Using tension sensors with distinct mechanical sensitivities to test force-based versus extension-based control of vinculin loading. (A-C) Representative images of the localization of a trio of vinculin tension sensors to FAs. (D) Normalized histograms of acceptor intensities at FAs are indistinguishable between the three sensors. (E-G) Representative images of masked FRET efficiency and (H) normalized histograms of average FA FRET reported by each sensor. (I-K) Representative images of forces and (L) normalized histograms of average vinculin force in FAs reported by each sensor. (M-O) Representative images of extension and (P) normalized histograms of average vinculin extension in FAs reported by each sensor. Note that ~12% of FAs exhibited negative forces/extensions and were excluded from the analysis in panels (L and P). All normalized histograms depict data from individual FAs; n = 98, 79, 63 cells and n = 11900, 9461, 7569 FAs for (GGSGGS)$_{5,7,9}$ extensible domains, respectively; data pooled from six independent experiments; ****p<0.0001, n.s. not significant (p≥0.05), ANOVA. See ***Supplementary file 4*** for exact p-values and multiple comparisons test details.

DOI: https://doi.org/10.7554/eLife.33927.024

The following figure supplements are available for figure 4:

**Figure supplement 1.** Schematic depiction of force-extension relationships and potential force- and extension-control paradigms.

*Figure 4 continued on next page*

*Figure 4 continued*

DOI: https://doi.org/10.7554/eLife.33927.025

**Figure supplement 2.** Evaluating gradients in vinculin extension and force at the sub-FA length scale.

DOI: https://doi.org/10.7554/eLife.33927.026

**Figure supplement 3.** Effect of Y-27632 treatment on vinculin extension-control.

DOI: https://doi.org/10.7554/eLife.33927.027

**Figure supplement 4.** Three versions of VinTS respond similarly to Y-27632 treatment.

DOI: https://doi.org/10.7554/eLife.33927.028

**Figure supplement 5.** Effect of disrupting vinculin-talin interactions on vinculin extension-control.

DOI: https://doi.org/10.7554/eLife.33927.029

**Figure supplement 6.** Three versions of VinTS-A50I exhibit similar FA morphologies, cell morphologies, and sensor localization to FAs.

DOI: https://doi.org/10.7554/eLife.33927.030

**Figure supplement 7.** Effect of substrate stiffness on vinculin extension-control.

DOI: https://doi.org/10.7554/eLife.33927.031

**Figure supplement 8.** Three versions of VinTS respond similarly to softer substrates.

DOI: https://doi.org/10.7554/eLife.33927.032

**Figure supplement 9.** Various structural arrangements within FAs could lead to force-based or extension-based control.

DOI: https://doi.org/10.7554/eLife.33927.033

model parameter corresponds to an alterable variable in sensor design ($R_0$ = FRET pair, $L_P$ = composition of extensible domain, $N$ = length of extensible domain), we can bypass the need to iteratively 'guess and check' the performance of new sensors, and, instead, predict the performance of unstructured polypeptide-based tension sensors *in silico*. Since our measurements and modeling efforts indicate that both force and extension might be key mechanical variables in different contexts, we report the predicted mechanical responses for simulated sensors in terms of both force and extension. The predicted relationships between force, extension, and FRET for a single sensor can be concisely described by three metrics as depicted in *Figure 5A–C*: (1) a FRET dynamic range ($\Delta FRET$), which is defined as the change in FRET efficiency from an unloaded state to an experimentally-determined 5% noise floor; (2) a target force ($F_{target}$), which indicates the midpoint of force range a sensor is functional, and is defined as $F_{target} = \Delta F/2$; and (3) a target extension ($r_{z,target}$), which is analogous to target force. Examining the predicted $\Delta FRET$, $F_{target}$, and $r_{z,target}$ for a variety of Clover-mRuby2 TSMods containing extensible domains of various lengths and compositions, we generate a 'roadmap' for future Clover-mRuby2 sensor design (*Figure 5D–F*, see *Supplementary file 1* for a list of reported polypeptide mechanical properties justifying the range over which simulations were performed). Additional roadmaps were generated for other commonly used FRET pairs (*Figure 5—figure supplement 1*). With these roadmaps as a guide, the rational design and implementation of future tension sensors with diverse and *a priori* specified properties is now possible.

## Discussion

Molecular tension sensors provide insight into the mechanical loading of individual proteins inside cells but have been limited by small dynamic ranges and an inability to tune their mechanical sensitivities. In this work, we leveraged the predictable mechanical responses of unstructured polypeptides to create and characterize a suite of TSMods with improved, specified, and tunable mechanical properties. These new modules were used to create a sensor optimized for the detection of loads across vinculin, as well as a suite of sensors that revealed an extension-control paradigm mediating vinculin loading. Additionally, we used the model to predict the mechanical response of over 1000 distinct sensors, enabling the rational design of future molecular tension sensors for diverse applications.

Through a systematic examination of the individual components of TSMods, we identified increased Förster distance, the use of FPs that lack unstructured residues, and extensible domains comprised of tunable unstructured polypeptides as key to the rational design of the next generation of tension sensors. Surprisingly, we observed context-dependent mechanical behaviors in TSMods, as responses for the same constructs were mechanically distinct in in vitro and *in cellulo* experiments. Differences in many aspects of these environments, including ionic strength, pH, and crowding,

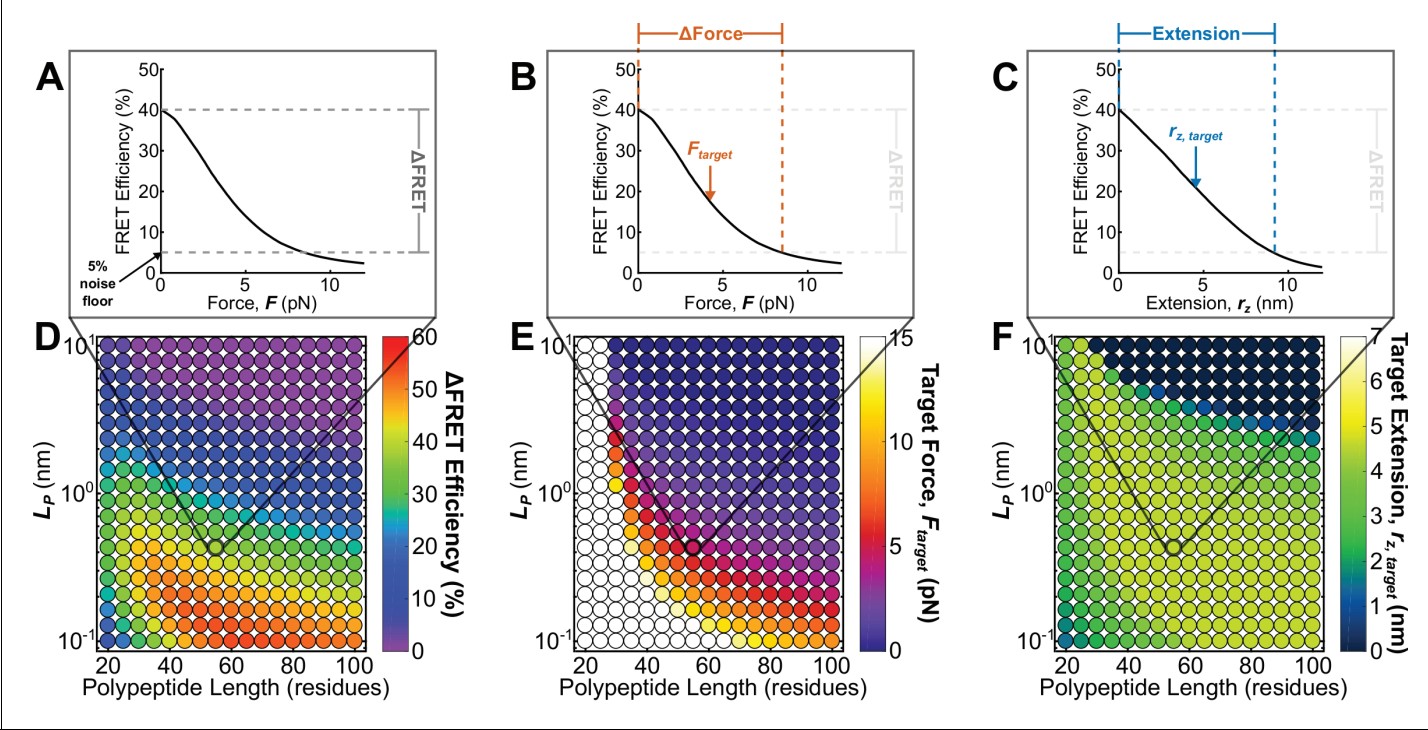

**Figure 5.** Roadmap to enable the rational design of FRET-based molecular tension sensors. (A-C) Representative plots of relationships between FRET efficiencies, forces, and extensions reported by a single sensor, highlighting $\Delta FRET$ (A), $\Delta Force$ (B), and polypeptide extension $r_z$ (C) required to stretch a sensor from its resting state to the 5% FRET noise floor. (D-F) Parameter space highlighting the predicted $\Delta FRET$ at the 5% noise floor (D) as well as the target force $F_{target}$ (E) and target extension $r_{z,target}$ (F) predicted for a variety of Clover-mRuby2-based sensors with a variety of polypeptide lengths (x-axis), and polypeptide mechanical properties (y-axis). Each point represents a single potential TSMod design.

DOI: https://doi.org/10.7554/eLife.33927.034

The following figure supplement is available for figure 5:

**Figure supplement 1.** Roadmaps to guide the rational design of FRET-based molecular tension sensors for some commonly used FRET-pairs.

DOI: https://doi.org/10.7554/eLife.33927.035

could explain the variety of reported mechanics. However, we surmise that crowding effects are less likely for two reasons. First, previous work has shown that crowding effects are more prominent at longer linker lengths (*Ohashi et al., 2007*). Additionally, in our in vitro and *in cellulo* systems, we observed that (GPGGA)$_n$-based polypeptides exhibit higher FRET in vitro, while (GGSGGS)$_n$ poly-peptides exhibited lower FRET in these conditions. These divergent behaviors are not readily explained by crowding effects, which should increase FRET in both constructs (*Ohashi et al., 2007*). While we do not establish the origin of these environmental factors in this work, these results raise potential concerns about the applicability of in vitro (single molecule-based) calibrations to sensors that are almost exclusively utilized *in cellulo*.

To circumvent the need for in vitro calibrations, we developed and validated a first-principles model which predicts TSMod mechanical sensitivity using *in cellulo* measurements of the mechanical properties of unstructured polypeptides used as the extensible domain. As quantitatively identical and physically meaningful parameters described the behavior of TSMods in both unloaded and loaded conditions, measurements of the *in cellulo* behavior of soluble TSMods can be used to predict the *in cellulo* behavior of tension sensors under load. In future tension sensor studies, this model could be paired with *in cellulo* measurements of $L_P$ to (1) help identify *in vitro* conditions that suitably match *in cellulo* systems, (2) re-calibrate sensors in new systems with distinct environmental properties (cell types, subcellular compartments), or (3) detect, and account for, changes in TSMod function following biological treatments or through time.

To demonstrate both the predictive power of the model and the improvements to these new *in cellulo* calibrated TSMods, we generated a variety of new vinculin tension sensors, the first of which was optimized to detect the 1–6 pN forces thought to be borne by vinculin (*Grashoff et al., 2010*). This optimized sensor exhibited a marked increase in the ability to detect asymmetric distributions of molecular tension within single FAs. Similar tension asymmetries have been observed external to the cell using both high resolution traction force microscopy (*Legant et al., 2013*; *Plotnikov et al., 2012*) and extracellular tension sensors (*Blakely et al., 2014*; *Morimatsu et al., 2015*). Intracellularly, gradients in vinculin tension have been reported in cells adhered to micropost arrays, although tension asymmetries were mainly attributed to the presence of discontinuous substrates (*Sarangi et al., 2017*). We show that these asymmetric molecular loads are detectable without the need for super-resolution imaging and are transmitted through vinculin even on continuous substrates.

Leveraging a suite of *in cellulo* calibrated vinculin tension sensors with distinct mechanical properties, we investigated a question that was previously technically inaccessible: are the forces across or extensions of proteins subject to cellular control? We observed exclusively extension-control in vinculin, and showed it is robust to a reduction in cell contractility, ablation of vinculin-talin interactions, and plating cells on substrates with physiologically-relevant stiffness. Simulations of FA structures suggested extension-control paradigms were likely due to discrete extensions from the cytoskeleton. Such discrete displacements could arise due to the step-like activities of molecular motors (myosin II) or actin polymerization, suggesting a critical role for the cytoskeleton in these processes. Furthermore, these structural simulations indicate that force-control and extension-control are sensitive to the relative abundance and stiffness of various proteins within the bulk structure. Thus, in addition to extension-controlled protein loading, it is possible that other load-bearing proteins might be subject to either type of control or that a specific protein in distinct cellular contexts could switch control modalities. A key question for future study is how proteins with multiple repeated domains that exhibit non-monotonic force-extension curves, such as talin (*Yao et al., 2016*) and filamin (*Furuike et al., 2001*), are mechanically regulated in cells.

The results of this study provide an updated picture of vinculin function at FAs. Specifically, once vinculin is activated through interactions with talin and/or actin (*Bakolitsa et al., 2004*; *Bois et al., 2006*; *Chen et al., 2006a*), it is pulled until a given extension is achieved, not a set force. Thus, the load-dependent maintenance of vinculin activation (*Dumbauld et al., 2013*) may be determined by vinculin extension, rather than force. More broadly, extension may be the most pertinent biophysical variable governing the initiation of vinculin-dependent and possibly other mechanosensitive signaling pathways. We note this picture is consistent with recent results that show VinTS FRET does not change on gels (*Kumar et al., 2016*) and that vinculin is still subject to mechanical load following the ablation of vinculin-talin interactions or partial ROCK-inhibition (*Rothenberg et al., 2018c*). Extrapolated to longer length scales, this extension-control paradigm agrees well with reports that cells induce similar strains within extracellular environments of differing stiffness (*Lin et al., 2018*; *Saez et al., 2005*) and that strain determines the activation of touch-sensitive channels in in vivo models (*Eastwood et al., 2015*).

To enable the rational design of future unstructured polypeptide-based sensors, we simulated the parameter space defined by the previously reported polypeptide mechanics ($0.1 < L_P < 10$ nm) and lengths (<100 residues) not likely to adversely affect protein function for several commonly used FRET-pairs. The prediction of more than 1000 possible tension sensor designs should allow for the creation of tension sensors suitable for most experiments in either extension- or force-based control paradigms. However, we note that measurements of force in an extension-control regime will be dependent on the stiffness of the sensor design. We speculate that this could be one possible reason for a variety of forces reported by different sensor designs for the same protein (*Erickson, 2017*).

In total, this work provides the biophysical foundation for understanding molecular tension sensor function and delivers a suite of *in cellulo*-calibrated sensors whose distinct and predictable mechanical sensitivities can be leveraged to gain unique molecular understanding of the role of mechanical forces and extensions in biological systems. These advancements should expedite deployment of molecular tension sensors in diverse biological contexts where mechanical cues and cellular force generation have long been thought to play critical, but unexplored, roles.

# Materials and methods

## Key resources table

| Reagent type (species) or resource | Designation | Source or reference | Identifiers | Additional information |
|---|---|---|---|---|
| Cell line (*Mus musculus*) | Vinculin -/- mouse embryonic fibroblast | PMID: 20181946 | NA | |
| Cell line (*Homo sapiens*) | HEK293, Human embryonic kidney cells | ATCC Cat# CRL-1573 | RRID:CVCL_0045 | |
| Transfected construct | pcDNA3.1(+) | Invitrogen, Carlsbad, CA | | |
| Transfected construct | VinculinTS | Addgene, Cambridge, MA; PMID: 20613844 | Plasmid #26019 | |
| Transfected construct | tCRMod-GGSGGS5 | Addgene, Cambridge, MA; this work | Plasmid #111760 | |
| Transfected construct | tCRMod-GGSGGS7 | Addgene, Cambridge, MA; this work | Plasmid #111761 | |
| Transfected construct | tCRMod-GGSGGS9 | Addgene, Cambridge, MA; this work | Plasmid #111762 | |
| Transfected construct | VinTS- tCRMod-GGSGGS5 | Addgene, Cambridge, MA; this work | Plasmid #111763 | |
| Transfected construct | VinTS- tCRMod-GGSGGS7 | Addgene, Cambridge, MA; this work | Plasmid #111764 | |
| Transfected construct | VinTS- tCRMod-GGSGGS9 | Addgene, Cambridge, MA; this work | Plasmid #111765 | |
| Sequence-based reagent | Oligonucleotides detailed in *Supplementary file 2* | this work | NA | |
| Chemical compound, drug | Y-27632 | Sigma Aldrich, St. Louis, MO | Y0503; PubChem Substance ID 24277699 | Used at 25 µM |
| Software, algorithm | ImageJ | US National Institutes of Health, Bethesda, MD | RRID:SCR_003070 | http://imagej.nih.gov/ij/ |
| Software, algorithm | Image Corrections | PMID: 25640429; doi.org/10.1007/s12195-015-0404-9 | NA | https://gitlab.oit.duke.edu/HoffmanLab-Public/image-preprocessing (*Rothenberg et al., 2018b*; copy archived at https://github.com/elifesciences-publications/HoffmanLab-image-preprocessing) |
| Software, algorithm | FRET calculations from 3-cube imaging | PMID: 16815904; doi.org/10.1007/s12195-015-0404-9 | NA | https://gitlab.oit.duke.edu/HoffmanLab-Public/fret-analysis (*Rothenberg et al., 2018a*; copy archived at https://github.com/elifesciences-publications/HoffmanLab-fret-analysis) |
| Software, algorithm | FRET calculations from spectrofluorometry | PMID: 16055154 | NA | https://gitlab.oit.duke.edu/HoffmanLab-Public/fluorimetry-fret (*LaCroix et al., 2018c*; copy archived at https://github.com/elifesciences-publications/HoffmanLab-fluorimetry-fret) |
| Software, algorithm | TSMod calibration model | this work | NA | https://gitlab.oit.duke.edu/HoffmanLab-Public/tsmod-calibration-model (*LaCroix et al., 2018b*; copy archived at https://github.com/elifesciences-publications/HoffmanLab-tsmod-calibration-model) |
| Software, algorithm | FA structural model | this work | NA | https://gitlab.oit.duke.edu/HoffmanLab-Public/FA-structural-model (*LaCroix and Hoffman, 2018a*; copy archived at https://github.com/elifesciences-publications/HoffmanLab-FA-structural-model) |

## Cell culture and transfection

Vinculin -/- MEFs (kindly provided by Dr. Ben Fabry and Dr. Wolfgang H. Goldmann (*Mierke et al., 2010*), Friedrich-Alexander-Universitat Erlangen-Nurnberg) were maintained in high-glucose DMEM with sodium pyruvate (D6429, Sigma Aldrich, St. Louis, MO) supplemented with 10% FBS (HyClone, Logan, UT), 1% v/v non-essential amino acids (Invitrogen, Carlsbad, CA), and 1% v/v antibiotic-antimycotic solution (Sigma Aldrich). Vinculin knockout was confirmed by western blot and immunofluorescent staining with mouse anti-vinculin antibody (V9131, Sigma Aldrich, dil. 1:5000, 1:500, respectively). Mycoplasma testing of this cell line by Duke Cell Culture Facility was negative. HEK293 cells were maintained in high-glucose DMEM (D5796, Sigma Aldrich) supplemented with 10% FBS (HyClone) and 1% v/v antibiotic-antimycotic solution (Sigma Aldrich). Cells were grown at 37°C in a humidified 5% $CO_2$ atmosphere. Cells were transfected at 50–75% confluence in 6-well tissue culture plates using Lipofectamine 2000 (Invitrogen) following the manufacturer's instructions.

## Generation of TSMods and vinculin tension sensor constructs

Constructs were created from the previously generated pcDNA3.1 mTFP1 (*Ai et al., 2006*; *Grashoff et al., 2010*) and pcDNA3.1 Venus (A206K) (*Grashoff et al., 2010*; *Nagai et al., 2002*) as well as pcDNA3.1 Clover (Addgene 40259) and pcDNA3.1 mRuby2 (Addgene 40260) (*Lam et al., 2012*). Minimal versions of single FPs were generated via Polymerase Chain Reaction (PCR) and inserted into pcDNA3.1 via EcoRI/NotI digestion and subsequent ligation (T4 DNA Ligase, NEB, Ipswich, MA). Specifically, creation of minimal FPs involved deletion of the 11 C-terminal residues in mTFP1 and Clover, and the first and second N-terminal residues in Venus and mRuby2 after the start codon. Oligonucleotide primers used to generate full-length and minimal versions of mTFP1, Venus A206K, Clover, and mRuby2 are detailed in *Supplementary file 2*.

The FP component fragments of the mTFP1-Venus and Clover-mRuby2 TSMods were derived from pcDNA3.1 TS module (*Grashoff et al., 2010*) and pcDNA3.1-Clover-mRuby2 (Addgene 49089) or the minimal FP variants described above. The extensible $(GPGGA)_n$ and $(GGSGGS)_n$ extensible domains were derived from pcDNA3.1 TS module (*Grashoff et al., 2010*) and pET29CLY9 (Addgene 21769) (*Evers et al., 2006*), respectively. Gibson assembly was used to construct TSMods containing a given FRET pair and extensible domain from three fragments: (1) vector backbone and donor FP (complementary regions: 5'-ampicillin gene, 3'-donor FP), (2) extensible domain region (complementary regions: 5'-donor FP, 3'-acceptor FP), and (3) vector backbone and acceptor FP (complementary regions: 5'-acceptor FP, 3'-ampicillin gene). Primers used to generate the extensible domain region in this reaction scheme were designed to nonspecifically target the repetitive extensible domain sequence, thereby generating extensible domains of various lengths. Following assembly and transformation into DH5α competent cells, single colonies were assayed for extensible domain length by DNA sequencing. Oligonucleotide primers used to generate TSMods are detailed in *Supplementary file 2*.

All variants of the vinculin tension sensor were derived from pcDNA3.1 Vinculin TS (*Grashoff et al., 2010*). In a cloning strategy analogous to that described above for the TSMods, Gibson assembly techniques were used to assemble vinculin tension sensors containing various minimal Clover-mRuby2 TSMods based on three fragments: (1) vector backbone and vinculin head domain residues 1–883 (complementary regions: 5'-ampicillin gene, 3'-Clover), (2) TSMod with desired $(GGSGGS)_n$ extensible domain (complementary regions: 5'-Clover, 3'-mRuby2), and (3) vector backbone and vinculin tail domain residues 884–1066 (complementary regions: 5'-mRuby2, 3'-ampicillin gene). Again, the assembled DNA fragments were transformed into DH5α competent cells and extensible domain length was verified for single colonies by DNA sequencing. Oligonucleotide primers used to generate vinculin tension sensors are detailed in *Supplementary file 2*. To generate A50I versions of the vinculin tension sensors, PCR was used to generate a fragment of the vinculin head domain containing the A50I mutation using forward primer 5'-AAT AAG CTT GCC ATG CCC GTC TTC CAC AC-3', reverse primer 5'-GCC GGA TCC GCA AGC CAG TTC-3', and template pEGFP-C1/GgVcl 1–851 A50I mutant (Addgene 46269). The product was insert into Clover-mRuby2-based vinculin tension sensors using 5'-HindIII/3'-BamHI. Plasmids will be distributed through Addgene (http://addgene.org).

## Cell seeding and preparation of glass and polyacrylamide substrates

For cell imaging on glass, no. 1.5 coverslips (Bioptechs, Butler, PA) placed in reusable metal dishes (Bioptechs) were coated overnight at 4°C with 10 µg/mL fibronectin (Fisher Scientific, Pittsburgh, PA). Transfected vinculin -/- MEFs expressing a given tension sensor construct were then trypsinized, transferred to the prepared glass-bottom dishes at a density 50,000 cells per dish, and allowed to spread to 4 hr in growth media. For fixed experiments, samples were then rinsed quickly with PBS, and fixed for 10 min with 3.7% methanol-free paraformaldehyde (Electron Microscopy Sciences, Hatfield, PA). For live experiments, growth media was exchanged, at least 1 hr before imaging, for imaging media - Medium 199 (Life Technologies, 11043) supplemented with 10% FBS (HyClone), 1% v/v non-essential amino acids (Invitrogen), and 1% v/v antibiotic-antimytotic solution (Sigma Aldrich). Live cell imaging was performed for up to 30 min at 37°C (Stable Z system, Bioptechs).

Polyacrylamide gels with elastic moduli of approximately 10 kPa (*Tse and Engler, 2010*) were created by mixing 10% acrylamide, 0.1% bis-acrylamide (BioRad, Hercules, CA) and 0.1% acrylic acid-NHS (Sigma Aldrich, to permit ECM functionalization), with polymerization initiated via addition of 0.1% ammonium persulfate and 0.05% N,N,N′,N′-tetramethylethylenediamine (Sigma Aldrich). Gels were cast between amino-silanated (*Tse and Engler, 2010*) and hydrophobic (Rain-X treated) coverslips (18 mm diameter, 40 µL gel solution per coverslip). Following gel polymerization (15 min), the top (hydrophobic) coverslip was removed, gels were rinsed thoroughly in HEPES buffer (50 mM, pH 8.5), then incubated overnight with fibronectin (10 µg/mL in HEPES buffer) at 4°C. ECM-coated gels were rinsed thoroughly with PBS prior to cell seeding. Transfected vinculin -/- MEFs expressing a given tension sensor construct were trypsinized, transferred to the ECM-coated gels at a density 50,000 cells per dish, and allowed to spread to 4 hr in growth media. Samples were then fixed for 10 min with 3.7% methanol-free paraformaldehyde (Electron Microscopy Sciences), then rinsed thoroughly in PBS. Finally, cells on gels were inverted onto bare no. 1.5 coverslips (Bioptechs) in reusable dishes (Bioptechs) and imaged.

## ROCK inhibitor (Y-27632) experiments

To inhibit Rho kinase (ROCK)-mediated myosin activity, cells were allowed to spread for 4 hr and treated with 25 µM Y-27632 (Sigma Aldrich), diluted from a 10 mM stock solution in deionized $H_2O$, 20 min before fixation. This treatment duration was the shortest capable of resulting in statistically significant loss of loading across vinculin (*Figure 4—figure supplement 3Q*), as has been shown in previous work (*Rothenberg et al., 2018c*).

## FRET imaging

All imaging was performed on an Olympus IX83 inverted epifluorescent microscope (Olympus, Center Valley, PA) equipped with a LambdaLS 300W ozone-free xenon bulb (Sutter Instruments, Novato, CA), a sCMOS ORCA-Flash4.0 V2 camera (Hamamatsu, Japan), motorized filter wheels (Sutter Instruments 10–3), and automated stage (H117EIX3, Prior Scientific, Rockland, MA). Image acquisition was controlled by MetaMorph Advanced software (Olympus). Samples were imaged at 60X magnification (Olympus, UPlanSApo 60X/NA1.35 objective, 108 nm/pix), using a three-image sensitized emission acquisition sequence (*Chen et al., 2006b*). The filter set for FRET imaging of mTFP1-Venus sensors includes mTFP1 excitation (ET450/30x, Chroma, Bellows Falls, VT), mTFP1 emission (Chroma, ET485/20 m), Venus excitation (Chroma, ET514/10x), and Venus emission (FF01-571/72, Semrock, Rochester, NY) filters, and a dichroic mirror (Chroma T450/514rpc). Images of mTFP1-Venus sensors were acquired in, sequentially, the acceptor channel (Venus excitation, Venus emission, 1000 ms exposure), FRET channel (mTFP1 excitation, Venus emission, 1500 ms exposure), and donor channel (mTFP1 excitation, mTFP1 emission, 1500 ms exposure). For Clover-mRuby2 sensors, we utilized the FITC and TRITC filters from the DA/FI/TR/Cy5−4 × 4 M-C Brightline Sedat filter set (Semrock), which provided efficient Clover excitation (FF02-485/20), Clover emission (FF01-525/30), mRuby2 excitation (FF01-560/25), and mRuby2 emission (FF01-607/36) filters, and appropriate dichroic mirror (FF410/504/582/669-Di01) for FRET imaging. Images of Clover-mRuby2 sensors were acquired in, sequentially, the acceptor channel (mRuby2 excitation, mRuby2 emission, 1500 ms exposure), FRET channel (Clover excitation, mRuby2 emission, 1500 ms exposure), and donor channel (Clover excitation, Clover emission, 1500 ms exposure).

## Quantitative FRET efficiency measurements from 3-cube FRET imaging

FRET was detected through measurement of sensitized emission (*Chen et al., 2006b*) and subsequent calculations were performed on a pixel-by-pixel basis using custom written code in MATLAB (Mathworks, Natick, MA) (https://gitlab.oit.duke.edu/HoffmanLab-Public/image-preprocessing) (*Rothenberg et al., 2015*; copy archived at https://github.com/elifesciences-publications/HoffmanLab-image-preprocessing). Prior to FRET calculations, all images were first corrected for uneven illumination, registered, and background-subtracted. Spectral bleed-through coefficients were determined through FRET-imaging of donor-only and acceptor-only samples (i.e. cells expressing a single donor or acceptor FP). Donor bleed-through coefficients (*dbt*) were calculated for mTFP1 and Clover as:

$$dbt = \left\langle \frac{I_f}{I_d} \right\rangle$$

where $I_f$ is the intensity in the FRET-channel, $I_d$ is the intensity in the donor-channel, and data were binned by donor-channel intensity. Similarly, acceptor bleed-through coefficients (*abt*) were calculated for Venus and mRuby2 as:

$$abt = \left\langle \frac{I_f}{I_a} \right\rangle$$

where $I_a$ is the intensity in the acceptor-channel, and data were binned by acceptor-channel intensity. To correct for spectral bleed-through in experimental data, corrected FRET images ($F_c$) were generated following the equation:

$$F_c = I_f - dbt * I_d - abt * I_a$$

After bleed-through correction, FRET efficiency was calculated following the equation:

$$E = \frac{I_d + \frac{F_c}{G}}{I_a}$$

where $G$ is a proportionality constant that describes the increase in acceptor intensity (due to sensitized emission) relative to the decrease in donor intensity (due to quenching) (*Chen et al., 2006b*). This constant depends on the specific FRET pair used, imaging system, and image acquisition settings, and was calculated for both mTFP1-Venus and Clover-mRuby2 biosensors through imaging donor-acceptor fusion constructs of differing but constant FRET efficiencies. See *Supplementary file 3* for bleed-through and $G$ coefficients.

Wherever possible, image analysis was standardized using custom-written Matlab software. Analysis parameters (*Supplementary file 3*) and thresholds for image segmentation were maintained across multiple days of experiments of the same type. For all TSMod and VinTS constructs, only cells with an average acceptor intensity within a pre-specified range were analyzed. This range was set to [1000 40000] for mTFP1-Venus-based sensors or [600 24000] for Clover-mRuby2-based sensors, resulting in exclusion of <10% of cells. Finally, for VinTS constructs, cells that were not fully spread were also excluded from analysis.

## Segmentation and analysis of VinTS in FAs

Post-processing of FRET images to segment and quantify the characteristics of individual FAs was performed using custom-written code in MATLAB (Mathworks). Briefly, FAs were identified and segmented on the acceptor channel using the water algorithm (*Zamir et al., 1999*). The resultant mask was applied across all images for ease of data visualization and quantification. For each identified FA, parameters describing its brightness in the acceptor channel, morphology, and molecular loading (FRET) were determined. To identify single cells, closed boundaries were drawn by the user based on the unmasked acceptor-channel image. From these cell outlines, parameters describing cell morphology and FA subcellular location were also determined.

Line scans of single FAs were performed using ImageJ software (US National Institutes of Health, Bethesda, MD; http://imagej.nih.gov/ij/). Specifically, the line tool was used to visualize the acceptor channel intensity profile across single, large FAs in the cell periphery. The coordinates of these lines, drawn axially starting from the tip of FAs distal to the cell body, were then transferred to masked

FRET efficiency images. Acceptor intensity and FRET efficiency profiles from single FAs were saved as text files for subsequent analysis.

## FRET efficiency calculations from spectrofluorometry

Hypotonic lysates were prepared from HEK293 cells as previously described (*Chen et al., 2005*). In addition to experimental samples, lysates from an equal number of untransfected cells were harvested and used as a reference background. Spectrofluorometric measurements were made with a Fluorolog-3 (FL3-22, HORIBA Scientific Jobin Yvon, Edison, NJ) spectrofluorometer with 1 nm step size, 0.2 s integration time, and 3 nm excitation and emission slit widths for all samples. For FRET measurements of mTFP1-Venus sensors, spectra were traced from 472 to 650 nm following donor excitation ($\lambda_{Dex}$) at 458 nm, and from 520 to 650 nm following acceptor excitation ($\lambda_{Aex}$) at 505 nm. For Clover-mRuby2 sensors, spectra were traced from 520 to 700 nm following donor excitation at 505 nm, and from 590 to 700 nm following acceptor excitation at 575 nm. The same settings were used to measure the emission spectra of full length and minimal FPs to confirm their spectral properties individually. Custom-written code in MATLAB (Mathworks) was used to calculate FRET efficiency via the (ratio)$_A$ method (*Majumdar et al., 2005*) as:

$$E = \frac{\varepsilon_A \lambda_{Aex}}{\varepsilon_D \lambda_{Dex}} \left( \frac{I_f \lambda_{Aem}}{I_a \lambda_{Aem}} - \frac{\varepsilon_A \lambda_{Dex}}{\varepsilon_A \lambda_{Aex}} \right)$$

where $I_f$ and $I_a$ are the intensities, at peak acceptor emission wavelength ($\lambda_{Aem}$, 530 nm for Venus, 610 nm for mRuby2), of the sample excited at donor and acceptor wavelengths, respectively. Donor and acceptor molar extinction coefficients ($\varepsilon_D$ and $\varepsilon_A$, respectively) were calculated from absorbance spectra measured on the same Fluorolog-3 spectrofluorometer in absorbance mode (1 nm step size, 0.1 s integration time, 2 nm excitation and emission slit widths) using previously-measured maximal extinction coefficients: 64,000 $M^{-1}cm^{-1}$ for mTFP1 (*Ai et al., 2006*), 93,000 $M^{-1}cm^{-1}$ for Venus (*Nagai et al., 2002*), 111,000 $M^{-1}cm^{-1}$ for Clover (), and 113,000 $M^{-1}cm^{-1}$ for mRuby2 (*Lam et al., 2012*). This approach was also used to confirm the absorbance spectra of the minimal FPs were unaltered as compared to the parent version.

## Statistics, bootstrapping, and data digitization

All statistical analyses, except numerical bootstrapping, were performed using JMP Pro 12 software (SAS, Cary, NC). ANOVAs were used to determine if statistically significant differences ($p<0.05$) were present between groups. If statistical differences were detected, Tukey's HSD post-hoc testing was used to perform multiple comparisons and assess statistical differences between individual groups (see *Supplementary file 4* for exact p-values and multiple comparisons test details). Box-and-whisker diagrams (*Figure 1F*, *Figure 1—figure supplement 2C*) display the following elements: center line, median; box limits, upper and lower quartiles; whiskers, 1.5x interquartile range; red filled circle, mean; open circles, outliers.

Numerical bootstrapping using the built-in Matlab (Mathworks) function *bootstrp.m* was used to calculate 95% confidence intervals for measurements of $L_P$. Specifically, for each of 200 bootstrapped samples, drawn with replacement from the pertinent dataset, the $L_P$ that best reflected that sample was calculated by chi-squared error minimization. Fluorescence-force spectroscopy data (*Brenner et al., 2016*) was digitized using the digitize2.m function in Matlab (Mathworks). To recapitulate the uncertainty in these published unloaded FRET and FRET-force datasets, random sets of 100 data points obeying a Gaussian distribution with the reported mean and standard deviation were used.

For FRET efficiency measurements, numerical bootstrapping of pilot data was used to determine the sample size required to estimate FRET efficiency to within 1% of the true population mean. This was determined to be 10–20 cells from three independent experiments for *in cellulo* measurements, or five independent samples for in vitro spectral FRET characterization. For fluorescent protein absorbance/emission spectra characterization, sample size was not pre-determined. Rather, the reproduction of data from independent experiments was deemed sufficient to draw conclusions about changes in the fluorescent protein spectral properties.

## Code availability

All code developed and utilized in this study is publically available at https://gitlab.oit.duke.edu/HoffmanLab-Public. Code used to measure and correct for chromatic aberration, uneven illumination, darkfield noise, and background intensities can be found in the 'image-preprocessing' repository https://gitlab.oit.duke.edu/HoffmanLab-Public/image-preprocessing (copy archived at https://github.com/elifesciences-publications/HoffmanLab-image-preprocessing). Code used to analyze FRET-based tension sensor data, including FRET corrections, object segmentation, object analysis, and cell segmentation can be found in the 'fret-analysis' repository https://gitlab.oit.duke.edu/HoffmanLab-Public/fret-analysis (copy archived at https://github.com/elifesciences-publications/HoffmanLab-fret-analysis). Code to perform spectral FRET analysis on in vitro fluorometric FRET experimental data can be found in the 'fluorimetry-fret' repository https://gitlab.oit.duke.edu/HoffmanLab-Public/fluorimetry-fret (copy archived at https://github.com/elifesciences-publications/HoffmanLab-fluorimetry-fret). Source code for the computational TSMod calibration model, which allows the user to simulate the mechanical response of molecular tension sensor modules, can be found in the 'tsmod-calibration-model' repository https://gitlab.oit.duke.edu/HoffmanLab-Public/tsmod-calibration-model (copy archived at https://github.com/elifesciences-publications/HoffmanLab-tsmod-calibration-model). Source code for the structural model of FA molecules, used to explore the physical limits of extension-control, can be found in the 'FA-structural-model' repository https://gitlab.oit.duke.edu/HoffmanLab-Public/FA-structural-model (copy archived at https://github.com/elifesciences-publications/HoffmanLab-FA-structural-model).

## Acknowledgements

The authors thank Dr. Ben Fabry and Dr. Wolfgang H Goldmann (Friedrich-Alexander-Universitat Erlangen-Nurnberg) for providing vinculin -/- cells, Dr. Chris Gilchrist for aid in polyacrylamide gel preparation, and Dr. Harold Erickson (Duke University) and Dr. Wendy Gordon (University of Minnesota) for comments and critical examination of the manuscript. This research was supported by the Searle Foundation, the March of Dimes Basil O'Conner Starter Scholar Award, the National Science Foundation CAREER #1454257, and the U.S. National Institutes of Health grants 1R21-HD084290-01 and F32-GM-119294.

## Additional information

### Funding

| Funder | Grant reference number | Author |
|---|---|---|
| March of Dimes Foundation | Basil O'Conner Starter Scholar Award | Brenton D Hoffman |
| National Science Foundation | CAREER #1454257 | Brenton D Hoffman |
| National Institutes of Health | 1R21-HD084290-01 | Brenton D Hoffman |
| National Institutes of Health | F32-GM-119294 | Matthew E Berginski |
| Searle Foundation | Searle Scholar Award | Brenton D Hoffman |

The funders had no role in study design, data collection and interpretation, or the decision to submit the work for publication.

### Author contributions

Andrew S LaCroix, Conceptualization, Resources, Software, Formal analysis, Investigation, Visualization, Methodology, Writing—original draft, Writing—review and editing, Generation and characterization of expression constructs, Image analysis software, Calibration model development, Structural model development; Andrew D Lynch, Software, Formal analysis, Investigation, Methodology, Calibration model development; Matthew E Berginski, Resources, Software, Funding acquisition, Investigation, Generation and characterization of expression constructs, Image analysis software; Brenton D Hoffman, Conceptualization, Software, Supervision, Funding acquisition, Methodology, Writing—original draft, Project administration, Writing—review and editing

## Author ORCIDs

Andrew S LaCroix (iD) https://orcid.org/0000-0003-0117-4785
Brenton D Hoffman (iD) http://orcid.org/0000-0003-2296-1331

## Decision letter and Author response

Decision letter https://doi.org/10.7554/eLife.33927.056
Author response https://doi.org/10.7554/eLife.33927.057

## Additional files

### Supplementary files

• Supplementary file 1. Persistence length ($L_P$) for a variety of published polypeptides. Overall, persistence lengths < 0.2 and > 5.0 nm are rarely observed. Synthetic homo-polymers (ex. poly-proline) can achieve larger persistence lengths.
DOI: https://doi.org/10.7554/eLife.33927.036

• Supplementary file 2. Primers used in this study.
DOI: https://doi.org/10.7554/eLife.33927.037

• Supplementary file 3. Spectral bleed-through coefficients and G factors for mTFP1-Venus and Clover-mRuby2 based tension sensor FRET efficiency calculations.
DOI: https://doi.org/10.7554/eLife.33927.038

• Supplementary file 4. Statistical test details including exact p-values for ANOVAs and post-hoc tests. Note, individual comparisons were made only when statistical significance was detected in ANOVAs.
DOI: https://doi.org/10.7554/eLife.33927.039

• Transparent reporting form
DOI: https://doi.org/10.7554/eLife.33927.040

### Data availability

Source data files have been provided for Figure 1G&H, Figure 2A&B, Figure 1-Figure supplement1A&B, Figure 1-Figure supplement 2A&B, Figure 1-Figure supplement 3B&C, Figure 2-Figure supplement2A&C, and Figure 2-Figure supplement 6. Due to their large size (>300GB), image datasets will be available from the corresponding author on reasonable request. Full sequence information for tCRMod-GGS5 (Plasmid #111760), tCRMod-GGS7 (Plasmid #111761), tCRMod-GGS9 (Plasmid#111762), VinTS-tCRMod-GGS5 (Plasmid #111763), VinTS-tCRMod-GGS7 (Plasmid #111764), andVinTS-tCRMod-GGS9 (Plasmid #111765) is available at Addgene (http://www.addgene.org). All software developed in the current study is publicly available on GitLab (https://gitlab.oit.duke.edu/HoffmanLab-Public), including code to perform image preprocessing ('image-preprocessing' repository), FRET analysis and FA segmentation ('fret-analysis' repository), fluorometric FRET analysis ('fluorimetry-fret' repository), calibration model simulations ('tsmod-calibration-model' repository), and FA structural model simulations ('FA-structural-model' repository).

The following datasets were generated:

| Author(s) | Year | Dataset title | Dataset URL | Database, license, and accessibility information |
|---|---|---|---|---|
| Brenton D Hoffman, Matthew E Berginski, Andrew S La-Croix | 2018 | tCRMod-GGSGGS9 | https://www.addgene.org/111762 | Publicly available at Addgene (iD 111762) |
| Brenton D Hoffman, Matthew E Berginski, Andrew S La-Croix | 2018 | tCRMod-GGSGGS7 | https://www.addgene.org/111761 | Publicly available at Addgene (ID 111761) |
| Brenton D Hoffman, Matthew E Bergins- | 2018 | tCRMod-GGSGGS5 | https://www.addgene.org/111760 | Publicly available at Addgene (ID 111760 |

| | | | | |
|---|---|---|---|---|
| ki, Andrew S La-Croix | | | | ) |
| Brenton D Hoffman, Matthew E Berginski, Andrew S La-Croix | 2018 | tCRMod-GGSGGS5 | https://www.addgene. org/111763 | Publicly available at Addgene (ID 111763) |
| Brenton D Hoffman, Matthew E Berginski, Andrew S La-Croix | 2018 | tCRMod-GGSGGS7 | https://www.addgene. org/111764 | Publicly available at Addgene (ID 111764) |
| Brenton D Hoffman, Matthew E Berginski, Andrew S La-Croix | 2018 | tCRMod-GGSGGS9 | https://www.addgene. org/111765 | Publicly available at Addgene (ID 111765) |

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

## Appendix 1

DOI: https://doi.org/10.7554/eLife.33927.041

# Development, validation, and implementation of a computational model describing the mechanical sensitivity of TSMods

In this appendix, we present a modeling-based approach to predict the relationship between applied force and measured FRET efficiency in a collection of genetically-encoded fluorescent protein (FP)-based molecular tension sensors that use unstructured polypeptides as the extensible domain (overview in Section I). This calibration model consists of a combination of accurate numerical predictions of biopolymer extension under load (Section II), simple empirical estimates correcting for the finite size (Section III) and dynamics of the FPs (Section IV), and established distance-FRET relationships (Section V). The effects of force on tension sensor module (TSMod) FRET efficiency are simulated using a Monte Carlo calculation scheme described in Section VI. To develop a modelling-based calibration approach for TSMods based on unstructured polypeptides, we investigated the ability of the proposed model to describe TSMod mechanical behaviors in both unloaded (Section VII) and loaded (Section VIII) conditions. Finally, we provide a comparison of the proposed model to other previous estimates of TSMod mechanical behavior (Section IX).

We demonstrate that the proposed model predicts the relationship between mechanical forces and FRET efficiency with sufficient accuracy to aid in the design of TSMods. Specifically, we show that the model can describe multiple experimental datasets with physically accurate parameters. Furthermore, calibrations suggest that vinculin tension sensors based on these TSMods report forces that are consistent with estimates from previously developed tension sensors (*Grashoff et al., 2010*) as well as those from traction force microscopy (*Balaban et al., 2001*).

## I. Overview of model

Estimates of TSMod mechanical sensitivity relate measured FRET efficiencies to applied forces. However, force and FRET are neither linearly nor simply related, as the relationship will depend on the mechanical properties of the extensible domain as well as the manner in which the extensible domain is connected to the fluorescent moiety. When considering TSMods with unstructured polypeptide extensible domains, there are two key distance metrics that must be considered. First, the end-to-end distance of the polypeptide ($r_e$) can be directly related to force via various mechanical models of polymer mechanics (Section II, *Figure 2–Figure supplement 1A*). Second, the chromophore separation distance ($r_c$) is directly related to FRET via established distance-FRET relationships including the Förster equation (Section IV, *Figure 2–Figure supplement 1C*). The relationship between $r_e$ and $r_c$ therefore is of critical importance and requires knowledge of the radii of the FPs (Section III, *Figure 2–Figure supplement 1B*). Details of how the calculations were performed are provided in Section V.

## II. Extensible domain mechanics are well-described by the worm-like chain model

To describe the mechanics of the unstructured polypeptides that serve as the extensible elements of the TSMods, we employ the worm-like chain (WLC) model. This model, widely used in the study of the mechanics of biopolymers, captures the mechanical characteristics of a polymer in two key length scales, the persistence length ($L_P$) and the contour length ($L_C$). The persistence length describes the stiffness of the polymer, and is defined as the length over which the tangent direction of the polymer remains correlated (*Marko and Siggia, 1995*). The contour length describes the fully extended length of the polymer, and is defined by the product of the monomer size ($b_0 = 0.38 \ \mathrm{nm}$ for amino acid chains [*Evers et al., 2006*]) and the

number of monomers in the chain ($N$). Most often, the WLC model is presented as an approximation of the predicted force-extension curve, relating the average extension of the polymer to an applied force (**Marko and Siggia, 1995**):

$$F_z = \frac{k_b T}{L_P} \left( \frac{1}{4} \left( 1 - \frac{\langle r_z \rangle}{L_C} \right)^{-2} - \frac{1}{4} + \frac{\langle r_z \rangle}{L_C} \right) \tag{1}$$

where $F_z$ is the force on the polymer chain, $\langle r_z \rangle$ is the average chain extension, $k_b$ is the Boltzmann constant, and $T$ is temperature. A more accurate force-extension relation was developed by (**Bouchiat et al., 1999**) by a simple seventh order polynomial correction scheme:

$$F_z = \frac{k_b T}{L_P} \left( \frac{1}{4} \left( 1 - \frac{\langle r_z \rangle}{L_C} \right)^{-2} - \frac{1}{4} + \frac{\langle r_z \rangle}{L_C} + \sum_{i=2}^{i \leq 7} \alpha_i \left( \frac{\langle r_z \rangle}{L_C} \right)^i \right) \tag{2}$$

where $\alpha_2 = -0.5164228$, $\alpha_3 = -2.737418$, $\alpha_4 = 16.07497$, $\alpha_5 = -38.87607$, $\alpha_6 = 39.49944$, and $\alpha_7 = -14.17718$.

However, despite the widespread and successful use of these approximations of the force-extension curve of a WLC in the quantification of biopolymer mechanics (see **Supplementary file 1**), these relationships are not directly compatible with most FRET-based measurements for several reasons. The issue most pertinent to this work is that **Equations 1 and 2** relate force to the average extension ($\langle r_z \rangle$) of the polypeptide chain, which is zero in the absence of force. However, the non-zero polypeptide end-to-end distance ($r_e$), which describes the finite rest length of the extensible domain, is more directly related to the chromophore separation distance ($r_c$), which is the key distance determining FRET efficiency ($E$). Additionally, the nonlinear relationships between these quantities ($r_e$, $r_c$, and $E$) prevent the use of simple heuristic correction schemes to relate various distance metrics (**Evers et al., 2006**).

Previous work has developed formalisms that enable these concerns to be accounted for (**Becker et al., 2010**; **Evers et al., 2006**; **Vogel et al., 2012**; **2014**), but they have yet to be applied to the development of FRET-based TSMods. A key realization is that TSMods are molecular-scale tools and therefore will be subject to thermal fluctuations (**Vogel et al., 2012**). Therefore, to begin it is necessary to describe the end-to-end distance of the extensible polypeptide ($r_e$) as a probability density function $P(r_e)$, which is obtained by integrating the end-to-end vector probability density function $Q(\vec{R})$ over all the angles through which the chain may rotate:

$$P(r_e) = \int_0^\pi \int_0^{2\pi} Q(\vec{R}) r_e^2 \sin \phi \, d\theta d\phi \tag{3}$$

yielding

$$P(r_e) = 4\pi r_e^2 Q(r_e) \tag{4}$$

Forces will alter this distribution following the widely-applied Boltzmann relation:

$$P(r_e, F_z) = P(r_e) \exp\left( \frac{F_z r_e}{k_b T} \right) \tag{5}$$

where $P(r_e, F_z)$ is the probability of observing a given end-to-end distance for the polymer chain at a given force $F_z$. The average extension of the polypeptide chain ($\langle r_z \rangle$), which is similarly affected by forces following the Boltzmann relation, was calculated as:

$$\langle r_z \rangle = \int_{-\infty}^{\infty} r_z Q(r_z) \exp\left( \frac{F_z r_e}{k_b T} \right) dr_z \tag{6}$$

However, as stated above, this parameter has limited relevance for further FRET calculations and was only calculated for comparison to classical WLC force-extension relationships (*Equations 1 and 2*).

A variety of closed-form equations that describe the behavior of biopolymers exist, but are accurate only in specific force regimes and ranges of polymer mechanics (*Becker et al., 2010*). For example, the second Daniels approximation (*Daniels, 1952*; *Yamakawa and Stockmayer, 1972*) describes $P(r_e)$ for small extensions of soft polymers ($L_C/L_P > 5$) following:

$$P(r_e) = 4\pi r_e^2 \left(\frac{3}{4\pi L_P L_C}\right)^{3/2} exp\left(\frac{-3r_e^2}{4L_P L_C}\right)$$

$$\left(1 - \frac{5L_P}{4L_C} + \frac{2r_e^2}{L_C^2} - \frac{33r_e^4}{80L_P L_C^3} - \frac{79L_P^2}{160L_C^2} - \frac{329r_e^2 L_P}{120L_C^3} + \frac{6799r_e^4}{1600L_C^4} - \frac{3441r_e^6}{2800L_P L_C^5} + \frac{1089r_e^8}{12800L_P^2 L_C^6}\right)$$

(7)

In other work Wilhelm and Frey develop an approximation that describes the behavior of short chain or stiff ($L_C/L_P < 2$) polymers (*Wilhelm and Frey, 1996*). Specifically Wilhelm and Frey provide two series expansions:

$$P(r_e) \propto \begin{cases} \frac{4\pi r_e^2 k_b T L_P}{2\pi L_C} \sum_{k=1}^{\infty} \pi^2 k^2 (-1)^{k+1} exp\left(-\frac{L_P}{L_C} k_b T \pi^2 k^2 \left(1 - \frac{r_e}{L_C}\right)\right) \\ \qquad where\ \frac{L_P}{L_C}\left(1 - \frac{r_e}{L_C}\right) \geq 0.2 \\ \\ \frac{4\pi r_e^2 L_P}{8\pi^{3/2} L_C} \sum_{k=1}^{\infty} \left(\frac{L_P}{L_C}\left(1 - \frac{r_e}{L_C}\right)\right)^{-3/2} exp\left(-\frac{L_C(k-0.5)^2}{L_P\left(1 - \frac{r_e}{L_C}\right)}\right) H_2\left[\frac{k-0.5}{\sqrt{\frac{L_P}{L_C}\left(1 - \frac{r_e}{L_C}\right)}}\right] \\ \qquad where\ \frac{L_P}{L_C}\left(1 - \frac{r_e}{L_C}\right) < 0.2 \end{cases}$$

(8)

Where $H_2$ is the second Hermite polynomial $H_2 = 4x^2 - 2$. However, these approximations were later found to be valid primarily at high extension (*Becker et al., 2010*).

Excitingly, Becker et al. developed an explicit expression which interpolates between a number of these approximations, including but not limited to those described above, and created an ansatz that accurately describes $P(r_e)$ over a wide range of polymer mechanics $0.05 < L_C/L_P < 50$ (*Becker et al., 2010*), and likely to much higher values in the absence of excluded volume effects. Normalizing the stiffness to the contour length such that $\kappa = L_P/L_C$, the final form of the interpolated probability density function $P(r_e)$ is given by:

$$P(r_e) = 4\pi r_e^2 J_{SYD}\left(\frac{1 - c(r_e/L_C)^2}{1 - (r_e/L_C)^2}\right)^{5/2} exp\left(\frac{\sum_{i=-1}^{0}\sum_{j=1}^{3} c_{i,j}\kappa^i (r_e/L_C)^{2j}}{1 - (r_e/L_C)^2}\right)$$

$$\times exp\left(-\frac{k\kappa ab(1+b)(r_e/L_C)^2}{1 - b^2(r_e/L_C)^2}\right) I_0\left(-\frac{k\kappa a(1+b)(r_e/L_C)}{1 - b^2(r_e/L_C)^2}\right)$$

(9)

where $J_{SYD}$ is the Shimada-Yamakawa J-factor:

$$J_{SYD} = \begin{cases} 112.04\kappa^2 e^{0.246/\kappa - a\kappa}, & \kappa > 1/8 \\ \left(\frac{3}{4\pi\kappa}\right)^{3/2}\left(1 - \frac{5\kappa}{4}\right), & \kappa \leq 1/8 \end{cases}$$

(10)

with constants $a = 14.054$, $b = 0.473$, and coefficients $c_{i,j}$ given as:

$$(c_{i,j})_{i,j} = \begin{bmatrix} -3/4 & 23/64 & -7/64 \\ -1/2 & 17/16 & -9/16 \end{bmatrix}$$

(11)

Additionally, $I_0$ is a modified Bessel function of the first kind and parameters $c$ and $d$ are defined as:

$$1 - c \cong \left(1 + \left(0.38\kappa^{-0.95}\right)^{-5}\right)^{-1/5}$$

(12)

$$1 - d \cong \begin{cases} 0, & \kappa < 1/8 \\ \frac{1}{0.177/(\kappa-0.111)+6.40(\kappa-0.111)^{0.783}}, & \kappa \geq 1/8 \end{cases} \tag{13}$$

To describe the behavior of the unstructured polypeptides used in the assembly of TSMod constructs, values of $L_C$, $L_P$ and $F_z$ were specified, and the inverse-transform-sampling method (**Titantah et al., 1999**) was used to generate sets of random numbers distributed according to $P(r_e, F_z)$ as specified by combining the Becker *et al.* ansatz (**Equation 9**) with the Boltzmann relation (**Equation 5**). Note that while this describes the mechanical behavior of the unstructured polypeptide, the overall force sensitivity and FRET output from a given TSMod construct will also depend on the radii and photophysical properties of the FPs attached to this extensible domain as well (discussed further in Sections III and IV).

To confirm that this formalism was both correctly implemented and suitable for the intended purposes, several calculations demonstrating the ability of our simulations to accurately reproduce previous results were completed. First, we verified the ability of the model to described the unloaded states of an unstructured polypeptide by comparing the zero-force state of the calculated $P(r_e)$ (**Equation 9**) to published simulations of a discretized version of the WLC model developed by Wilhelm & Frey that is often used for stiffer polymers (**Wilhelm and Frey, 1996**), and the second Daniels approximation for softer polymers (**Daniels, 1952**; **Yamakawa and Stockmayer, 1972**), observing excellent agreement in all cases (**Figure 2–Figure supplement 2A**). Next, we determined the predictions of the scaling between the average end-to-end distance ($\langle r_e \rangle$) and polypeptide length ($N$) for the proposed model, observing the classical $\langle r_e \rangle \sim N^{0.5}$ behavior characteristic of random chains (**Figure 2–Figure supplement 2B**). Finally, the model was used to generate force-extension curves (**Equation 6**) for various ratios of contour length to persistence length and compared to the most accurate numerical approximation of the force-extension curve of a WLC (**Bouchiat et al., 1999**) (**Equation 2**). The model agrees well with the numerical approximation across all $L_C/L_P$ ratios (**Figure 2–Figure supplement 2C**). Together these data show that this approach accurately simulates $P(r_e, F_z)$ for the expected mechanics (**Supplementary file 1**) and lengths achievable for the extensible domain of polypeptide-based TSMods.

## III. Effects of fluorescent proteins

Next, we sought a method to account for the finite size of the FPs attached to the ends of the extensible domain. Like any FRET-based biosensor, the pertinent energy transfer distance for a TSMod is the inter-chromophore distance ($r_c$), not simply $r_e$. Since the FPs have finite size, the difference between these two quantities is not negligible and has been approximated by several methods (**Brenner et al., 2016**; **Evers et al., 2006**). In light of the experimental complexities associated with single molecule fluorescence force spectroscopy, Brenner *et al.* utilized a simple approach and approximated $r_c = r_e + C$ (**Brenner et al., 2016**). However, it is generally recognized that FPs are free to rotate around the ends of the polypeptide in a random fashion requiring more complex relationships to relate $r_c$ to $r_e$. Attempts by Evers *et al.* to recapitulate this behavior by simulating 66 possible orientations for each FP at a given $r_e$ (4356 total conformations possible) and considering the average as an appropriate estimate of $r_c$ are a reasonable means of estimating these effects (**Evers et al., 2006**). However, Evers' approach does not yield a realistic limit at short polypeptide lengths. Specifically, as the polypeptide length approaches zero, this approach allows for the centers of the FPs to be closer than the sum of their individual radii (**Figure 2–Figure supplement 2D**). To address this issue and approximate the effects of steric hindrance between the two FPs, we used a heuristic approach involving a quadrature sum of the key distances in this problem:

$$r_c = \sqrt{(R_{FP1} + R_{FP2})^2 + r_e^2} \tag{14}$$

where $R_{FP1}$ and $R_{FP2}$ are the radii of the donor and acceptor FPs, respectively. Based on crystal structures of FPs (**Ai et al., 2006**; **Ormö et al., 1996**; **Rekas et al., 2002**) and estimates of FP hydrodynamic radii from fluorescence correlation spectroscopy (**Hink et al., 2000**), we set

$R_{FP1} = R_{FP2} = 2.3$ nm. For cyanine dyes, these quantities were set to $R_{FP1} = R_{FP2} = 0.95$ nm, which is mathematically equivalent to the reported hydrodynamic radii of Cy3 and Cy5, which are 0.90 nm (**Muddana et al., 2009**) and 1.00 nm (**Widengren and Schwille, 2000**), respectively. This steric hindrance approximation recapitulates the behavior of Evers' spherical integration approach for longer polypeptides while also providing a realistic limit for very short polypeptides (**Figure 2–Figure supplement 2D**). As the operation is always the same for a given separation distance and size of FPs, a transfer function (**Equation 14**) was written to a file and treated as a lookup table for the conversion of $r_e$ to $r_c$.

## IV. Accounting for rotational entropy phenomenologically

When applying load to a molecule, a portion of energy works against rotational entropy before the molecule is extended. This phenomenon is known to limit resolution in force spectroscopy measurements (**Neuman and Nagy, 2008**), and was therefore considered in our modeling of force sensitive biosensors. Previous descriptions of this phenomenon (**Brenner et al., 2016**) have defined a critical force:

$$F_{critial} = \frac{k_b T}{r_{c,0}}$$

(15)

which is required to orient the molecule in the direction of pulling, and is set by the thermal energy of the system and the unloaded distance between the FP chromophores ($r_{c,0}$). Thus, forces below $F_{critical}$ do not entirely act to increase the separation distance between the FPs. We heuristically account for these effects using:

$$|F| = \sqrt{F_z^2 + F_{critical}^2} - F_{critical}$$

(16)

This results in only a fraction of the total amount of force applied to the TSMod, $|F|$, leading to extension of the TSMod through the force impinging on the extensible domain itself $F_z$ (**Figure 2–Figure supplement 2E**). This causes the emergence of a regime of relative insensitivity of the sensors at very small forces (**Figure 2–Figure supplement 2F**). Notably, this simple equation recapitulates observations in single molecule force spectroscopy experiments (**Brenner et al., 2016**; **Grashoff et al., 2010**).

## V. Förster resonance energy transfer

FRET efficiency ($E$) was calculated for each $r_c$ in the set using either the Förster equation:

$$r_c = R_0 \sqrt[6]{\frac{(1-E)}{E}}$$

(17)

which assumes a dynamic isotropic regime for fluorescent moiety rotation ($\kappa^2 = 2/3$) and is applicable to only quickly rotating, small fluorescent moieties (e.g. cyanine dyes), or an alternative distance-FRET relationship:

$$r_c = R_0 \sqrt[6]{\frac{(1-E)^2}{E}}$$

(18)

which is valid in the static isotropic regime and more accurately describes energy transfer between fluorescent moieties (e.g. FPs) which rotate on timescales longer than those associated with FRET (**Vogel et al., 2012**; **2014**). The Förster radius ($R_0$) appears in both relationships and is defined as the distance at which a pair of fluorescent moieties exhibit 50% FRET efficiency. The value for $R_0$ was set, based on previous measurements, to 5.7 nm for the mTFP1-Venus FRET pair (**Ai et al., 2006**), 6.3 nm for the Clover-mRuby2 FRET pair (**Lam et al., 2012**), and 5.4 nm for the Cy3-Cy5 FRET pair (**Buckhout-White et al., 2014**; **Sanborn et al., 2007**). As before, either transfer function was written to a file and treated as a lookup table

for the conversion of any $r_c$ to $E$. The mean of the resultant FRET distribution was then reported as the FRET efficiency at a given force.

## VI. Calculations

Monte Carlo simulations were used to determine the average polypeptide extension ($\langle r_z \rangle$) as well as probability distributions for the polypeptide end-to-end distance ($r_e$), FP chromophore separation distance ($r_c$), and FRET efficiency ($E$) at a given force on the polypeptide ($F_z$), persistence length ($L_P$), and number of residues in the polypeptide ($N$, used to calculate contour length, $L_C = N * b_0$). First, a set of at least 10,000 random numbers obeying the probability distribution function $P(r_e)$ described by the Becker et al. ansatz (**Equation 9**) was generated from a uniform distribution using the inverse-transform sampling method (**Titantah et al., 1999**). Then, the impact of force on $P(r_e)$ was accounted for following the Boltzmann relation (**Equation 5**). Next, $P(r_e)$ was first converted to $P(r_c)$ using the $r_e$-to-$r_c$ lookup table described by **Equation 14**, and then $P(E)$ using the $r_c$-to-$E$ lookup table described by **Equation 17** for TSMod-like constructs with cyanine dyes as the fluorescent moiety or **Equation 18** for FP-based TSMods. Additionally, the force affecting the extension of the sensor ($|F|$) was determined from $F_z$ following **Equation 16**. Finally, the average of each population distribution as well as $|F|$, $N$, and $L_P$ were written to a line of a TSMod calibration text file. These operations were repeated from 0 to 50 pN of force at increments of 0.1 pN to map out the FRET efficiency ($E$) as a function of force ($|F|$) and extension ($r_z$) with a given set of parameters.

## VII. Model validation and estimates of persistence length in unloaded conditions

To evaluate whether the model can be used to describe the behavior of TSMods in unloaded conditions and provide quantitative measurements of $L_P$ from *in cellulo* measurements, a variety of experimental results were evaluated. First, we investigated whether the model accurately describes the relationship between FRET efficiency and polypeptide length for Clover-mRuby2 TSMods and provide reasonable estimates of $L_P$ for (GGSGGS)$_n$ or (GPGGA)$_n$ extensible domains in both in vitro and *in cellulo* environments. To limit the parameter space, estimates for the other two input parameters ($R_0$ and $R_{FP}$) were obtained from previous reports of the Förster radius of Clover and mRuby2 ($R_0 = 6.3$ nm[**Lam et al., 2012**]) and the hydrodynamic radii of GFP family FPs ($R_{FP} = 2.3$ nm[**Hink et al., 2000**]). This leaves $L_P$ as the sole adjustable parameter. Simulations were then used to predict the relationship between FRET efficiency and polypeptide length for a variety of polypeptide chains ($10 < N < 100$ residues) and persistence lengths ($0.1 < L_P < 2.0$ nm, 0.01 nm increments) and compared to in vitro FRET efficiency measurements of TSMods in cell lysates (**Figure 1G**). Model fits and bootstrapped 95% confidence intervals indicate a quantitative difference in the mechanics of (GPGGA)$_n$ ($L_P = 0.74 \pm 0.05$ nm) and (GGSGGS)$_n$ ($L_P = 0.33 \pm 0.05$ nm) extensible domains in vitro. Due to the presence of proline residues in the repeated structure of the (GPGGA)$_n$ extensible domain, this stiffer mechanical response is not unexpected (**Evers et al., 2006**). In contrast, fits to sensitized emission-based FRET efficiency measurements of sensors expressed *in cellulo* revealed quantitatively indistinguishable FRET-polypeptide length relationships for TSMods containing (GPGGA)$_n$ ($L_P = 0.50 \pm 0.02$ nm) and (GGSGGS)$_n$ ($L_P = 0.48 \pm 0.05$ nm) extensible domains (**Figure 1H**). These values are in rough agreement with previous reports of both spider silk (comprised of (GPGGA)$_n$ motifs, $L_P = 0.3 - 0.5$ nm) and (GGSGGS)$_n$ ($L_P = 0.45$ nm) mechanics (**Evers et al., 2006**; **Becker et al., 2003**). However, as different models of polymer mechanics as well as means of converting between FRET efficiency and fluorophore separation distance were used in the various studies, we hesitate to further interpret slight differences in these values. In total, these results confirm that there is a quantitative difference in extensible domain mechanics between in vitro and *in cellulo* environments, although less-so for (GGSGGS)$_n$-based TSMods, and provide quantitative measurements of $L_P$ that can be used to generate *in cellulo* TSMod calibrations.

To rule out the possibility that constraining $R_{FP}$ and/or $R_0$ to their published values was skewing estimates of $L_P$, we examined whether improved fits to experimental data could be obtained if either $R_{FP}$ or $R_0$ (in addition to $L_P$) were left unconstrained. Specifically, we compared simulated FRET-polypeptide length relationships for various combinations of either $L_P$ and $R_0$ (at $R_{FP} = 2.3$ nm) or $L_P$ and $R_{FP}$ (at $R_0 = 6.3$ nm) to experimental FRET-length measurements made in vitro (**Figure 2–Figure supplement 3**) or *in cellulo* (**Figure 2–Figure supplement 4**). Simulations included $L_P$ ranging from 0.10 to 1.00 nm, $R_{FP}$ from 1.7 to 3.4 nm, and $R_0$ from 5.1 to 6.8 nm. We then evaluated the chi-squared error between model predictions and experimental data for each such combination of $L_P$ and $R_{FP}$ (**Figure 2–Figure supplement 3A,B** and **4A,B**) or $L_P$ and $R_0$ (**Figure 2–Figure supplement 3C,D** and **4C,D**). In all cases, regardless of measurements made in vitro or *in cellulo*, minimal chi-squared error between model and experiment occurs at or close to the literature estimates of both $R_{FP}$ and $R_0$ (highlighted with vertical green rectangles). Finally, compared to the single unconstrained parameter $L_P$, leaving two parameters unconstrained leads to no significant improvement in fits to the relationship between FRET efficiency and polypeptide length in unloaded conditions (**Figure 2—figure supplement 3E,F** and **4E,F**). Thus, we conclude the use of literature-based estimates for key parameters did not bias our measurements of $L_P$.

## VIII. Model validation in loaded conditions

To evaluate the ability of the calibration model to describe the mechanical sensitivity of TSMods subject to mechanical loads, we examined model fits to previously published fluorescence-force spectroscopy data of (GPGGA)$_n$ polypeptides labelled with Cy3 and Cy5 dyes (**Brenner et al., 2016**). These TSMod-like constructs differ from the FP-based TSMods in terms of the physical size and the Förster radius of the fluorescent moiety. As such, we set $r_{FP1}$ = 0.9 nm and $r_{FP2}$ = 1.0 nm based on measurements of the hydrodynamic radii of Cy3 (**Muddana et al., 2009**) and Cy5 (**Widengren and Schwille, 2000**) dyes, respectively. We also set $R_0 = 5.4$ nm based on the reported photophysical properties of the Cy3-Cy5 FRET pair (**Buckhout-White et al., 2014**; **Sanborn et al., 2007**). As above, the one unconstrained parameter remaining, persistence length ($L_P$), describes the mechanical response of the extensible domain. Simulations of Cy3-Cy5 constructs with various polypeptide lengths ($10 < N < 100$ residues) and persistence lengths ($0.1 < L_P < 2.0$ nm) were then compared to experimentally measured FRET-polypeptide length and FRET-force relationships (**Figure 2**). The simulations where $L_P \sim 1.05$ nm agree well with both the FRET-polypeptide length relationship over the range of polypeptide lengths assessed (**Figure 2A**), as well as the measured relationship between FRET efficiency and force (**Figure 2B**). For comparison, additional FRET-polypeptide length and FRET-force relationships for a range of $L_P$ values between 1.0 and 1.15 nm are shown (lines in **Figure 2A**, shaded region in **Figure 2B**). Together, these data indicate that the proposed model can describe published fluorescence-force measurements of TSMod-like constructs in both unloaded and loaded conditions with identical parameters.

The relatively stiff mechanical response of the (GPGGA)$_n$ extensible domain ($L_P \sim 1.0$ nm) is somewhat unexpected given reports of other unstructured polypeptide mechanics (**Supplementary file 1**) and our own measurements of extensible domain mechanics in vitro (**Figure 1G**). However, these differences could be explained by the presence of stiff elements (chemical cross-linkers, double stranded DNA) used in the single molecule experimental setup as well as slightly different buffer conditions between the various experiments. To rule out the possibility that constraining $R_{FP}$ and/or $R_0$ to their published values was skewing estimates of $L_P$, we examined whether improved fits to experimental data could be obtained if either $R_{FP}$ or $R_0$ (in addition to $L_P$) were left unconstrained. Specifically, we compared simulated FRET-polypeptide length and FRET-force relationships for various combinations of either $L_P$ and $R_0$ (at $\langle R_{FP} \rangle = 0.95$ nm) or $L_P$ and $R_{FP}$ (at $R_0 = 5.4$ nm) to experimental FRET-length and FRET-force measurements (**Figure 2–Figure supplement 5**). Simulations included $L_P$ ranging from 0.20 to 1.25 nm, $R_{FP}$ from 0.3 to 2.0 nm, and $R_0$ from 4.8 to 6.5 nm. We then evaluated the chi-squared error between model predictions and experimental data for each such

combination of $L_P$ and $R_{FP}$ (**Figure 2–Figure supplement 5A,B**) or $L_P$ and $R_0$ (**Figure 2–Figure supplement 5C,D**). Minimal chi-squared error between model and experiment occurs at or close to the literature estimates of both $R_{FP}$ and $R_0$ (highlighted with vertical green rectangles). Finally, compared to the single unconstrained parameter $L_P$, leaving two parameters unconstrained leads to no significant improvement in fits to either the unloaded (**Figure 2–Figure supplement 5E**) or loaded (**Figure 2–Figure supplement 5F**) single molecule data. Thus, we find it unlikely that these stiff mechanics are an artifact of model parameter selection, and instead attribute the high $L_P$ to the experimental conditions and stiff covalent attachments (**Brenner et al., 2016**) required to connect different components in single molecule fluorescence force experiments. Specifically, the incorporation of an SMCC linker connecting the fluorescent molecules to the polypeptide chains likely contributes to the overall mechanics of the chain. The effects of this contribution may be greater for shorter polypeptide sequences, potentially explaining the increased discrepancy between the model and the data in this case.

## IX. Comparison to other descriptions of TSMod mechanical sensitivity

In validating the proposed model, we evaluated its ability to recapitulate published fluorescence-force spectroscopy measurements from (**Brenner et al., 2016**) of the mechanical response of $(GPGGA)_n$ extensible domains flanked by Cy3 and Cy5 fluorescent dyes (Section VIII). Based on their modelling efforts, one of the main conclusions of Brenner *et al.* was that $(GPGGA)_n$ behaves as a linear nanospring, not an unstructured polypeptide. This was primarily based on the observation of a linear relationship between average polypeptide end-to-end distance $\langle r_e \rangle$ and polypeptide length $N$, $\langle r_e \rangle \sim N^1$. On the contrary, for unstructured polypeptides, the predicted scaling is $\langle r_e \rangle \sim N^{1/2}$ (**Dittmore et al., 2011**) or $\langle r_e \rangle \sim N^{3/5}$ if excluded volume effects are considered (**Pincus, 1976**). Notably, the calibration model used in this work exhibits $\langle r_e \rangle \sim N^{1/2}$ scaling. Two main assumptions differentiated the Brenner *et al.* calculations from the proposed calibration model. First, the calibration model considers a probability distribution of sensor conformations while Brenner *et al.* performed distance-FRET conversions on an average value. As previously discussed in Section I, premature numerical averaging could lead to numerical artifacts when nonlinear transformations, as are present in distance-FRET conversions, are involved. Secondly, different assumptions were used pertaining to the connectivity of the fluorescent moieties to the polypeptide. Brenner *et al.* used a simple subtraction method which assumes that the fluorescent moiety is located at a constant offset from the extensible domain and $r_e = r_c - C$. This simple method has previously been reported to lead to numerical artifacts (**Evers et al., 2006**). In contrast, we used a heuristic $r_e$ to $r_c$ conversion (details in Section III) based on previous simulations of the various physical orientations of TSMod components (**Evers et al., 2006**).

To determine which of these distinct descriptions is most accurate, we investigated the effects of the assumptions in the two modelling approaches on the predictions of the scaling between $\langle r_e \rangle$ and $N$. We first examined what scaling is observed when the proposed calibration model is used to calculate the relationship between $\langle r_e \rangle$ and $N$ from experimental data. Starting with the published unloaded FRET efficiency measurements of Brenner *et al.* (**Figure 2A**), the calibration model was used to perform the abovementioned distance-FRET conversions and thus calculate $\langle r_e \rangle$. Plotting this data as a function of $N$ yielded a $\langle r_e \rangle \sim N^{0.57}$ scaling (**Figure 2–Figure supplement 6A**). While we are hesitant to over-interpret this result as it infers power-law relationships from data that varies by less than an order of magnitude, it nonetheless supports the notion that $(GPGGA)_n$ mechanics are consistent with unstructured polypeptides. Next, we investigated the prediction of the observed scaling between $\langle r_e \rangle$ and $N$ entirely *in silico*. When the calibration model is used to simulate $\langle E \rangle$ for a collection of TSMods of various lengths $N$, and subsequently the approach of Brenner *et al.* is used to convert these $\langle E \rangle$ to $\langle r_e \rangle$, we then observe an apparent $\langle r_e \rangle \sim N^1$ scaling over the range of polypeptide lengths examined ($25 < N < 50$ residues, **Figure 2–Figure supplement 6B**). Note that the polymer mechanics model used as the basis of the calibration model predicts

$\langle r_e \rangle \sim N^{0.5}$ (**Figure 2–Figure supplement 2B**). Thus, we conclude that the observation of a spring-like $\langle r_e \rangle \sim N^1$ behavior is a consequence of the assumptions made by Brenner et al., which have previously been shown to lead to poor estimates of $\langle r_e \rangle$ (**Evers et al., 2006**; **Vogel et al., 2012**). In total, these simulations and data show that the calibration model developed in this work is capable of accurately describing the mechanical properties of unstructured polypeptides, and suggests that extensible domains based on the consensus sequence (GPGGA)$_n$ fall within this class.

## Appendix 2

DOI: https://doi.org/10.7554/eLife.33927.042

### Examining extension-based control of vinculin loading with a structural model of a FA

Experiments that showed extension-based control of vinculin loading (**Figure 4**) leveraged three tension sensors with unique force-extension relationships, which were dictated by the unique mechanical responses of the extensible $(GGSGGS)_n$ polypeptide domains. Seeking to gain a better understanding of what kinds of physical interactions might give rise to force-controlled versus extension-controlled loading of different FA components, we developed a simple structural model of a single FA. The FA structural model consists of 170 elements each of which is described as a Hookean spring and can be conceptualized as either a 'sensor' element or an alternative 'linker' element. Based on the reported stratified organization of layers of proteins within FAs (**Kanchanawong et al., 2010**), we arranged the two elements in two parallel layers (**Figure 4—figure supplement 9A**). However, even with this simplified geometry, numerous scenarios are possible. Key variables include the relative stiffness and relative abundance of each element, as well as whether a bulk force input or a bulk extension input is provided to the structure. Therefore, the focus of these modeling efforts was to determine how the force-controlled or extension-controlled loading of individual FA proteins might be impacted by each of these variables in a variety of scenarios.

To examine a single possible physical scenario, structural models were considered in groups of three (**Figure 4—figure supplement 9B**), since experimental evaluation of force-based versus extension-based control similarly involved three sensors with distinct mechanical properties (**Figure 4**). Within each grouping of three, the stiffness of the linker elements, $k_L$, is held constant and only the stiffness of the sensor element was varied ($k_{S1}$, $k_{S2}$, or $k_{S3}$). The stiffness values used for $k_{S1}$, $k_{S2}$, and $k_{S3}$ were selected to approximate the relative differences in stiffness between the $(GGSGGS)_{5,7,9}$ extensible domains used in experiments and were estimated from linear fits to their predicted force-extension curves (**Figure 4—figure supplement 1**). **Figure 4—figure supplement 9B** depicts a single scenario in which equal numbers of sensor and linker elements ($N_S = N_L$) with comparable average stiffness ($\langle k_{Sj} \rangle = k_L$) are loaded by a bulk extension input ($\delta_0$) as an illustrative example.

Within any particular geometry, we evaluated the extent of force-controlled versus extension-controlled loading of the three distinct sensor elements. Following bulk loading of the structure, which in the illustrative case corresponds to a constant extension $\delta_0$, we solve for the extension of and force across the sensor elements in each of the three assemblies ($\delta_{Sj}$ and $F_{Sj}$, respectively). The extent of extension-controlled versus force-controlled mechanical behaviors is then evaluated by comparing the coefficient of variations, which is defined as standard deviation divided by the mean and abbreviated *CV* here, of the forces $CV(F_{Sj})$ and extensions $CV(\delta_{Sj})$ experienced by these three different stiffness sensor elements. A control metric quantitatively relating the two magnitudes of variation is defined as the $\log_2$ ratio of the two coefficients of variation:

$$Control\ Metric = \log_2 \left[ \frac{CV(F_{Sj})}{CV(\delta_{Sj})} \right] \tag{1}$$

In an extension-controlled assembly the forces will vary more than the extensions $CV(F_{Sj}) > CV(\delta_{Sj})$, resulting in a large and positive control metric, while the opposite will be true for a force-controlled assembly where $CV(F_{Sj}) < CV(\delta_{Sj})$. In the balanced scenario depicted in **Figure 4—figure supplement 9B**, which contains equal numbers of sensor and linker elements of identical average stiffness, we observe neither extension-control nor force-control following a bulk extension input to the assemblies. Thus, the control metric is close to zero.

In this framework, we investigated the effects of both bulk force ($F_0$) and extension ($\delta_0$) inputs to a variety of stratified structures comprised of variable abundances and stiffnesses of the two types of springs (*Figure 4—figure supplement 9C*). To accomplish this, the above calculations were repeated while the relative number of springs and $k_L$ were varied. As we are most interested in relative rather than absolute numbers of elements and element stiffness, the relative numbers and mechanics of the sensor and linker elements are defined as:

$$Relative\ Abundance = \frac{N_S}{N_L} \tag{2}$$

$$Relative\ Stiffness = \frac{\langle k_S \rangle}{k_L} \tag{3}$$

The number of sensor and linker elements ($N_S$ and $N_L$, respectively) and the stiffness of the linker element ($k_L$) were set such that the relative abundance and relative stiffness of the two elements varied between $2^{-4}$ and $2^4$. Note that when calculating the relative stiffness, the three distinct spring constants for the three sensor elements were averaged together. This definition is appropriate as the variation of the stiffness of the sensors springs ($k_{Sj}$) is significantly smaller (~30% variation) than the range of $k_L$ that was evaluated. These normalized parameters also allow us to draw conclusions that are independent of the absolute values of $k$ and $N$ that are simulated.

With a bulk force input ($F_0$), simple relationships that obey Hooke's Law are observed, and $F_0$ is evenly distributed across each sensor element:

$$F_{Sj} = F_0 / N_S \tag{4}$$

where $F_{Sj}$ refers to force observed in one of three group models. Subsequently, the force $F_{Sj}$ dictates the extension $\delta_{Sj}$ of individual sensor elements following:

$$\delta_{Sj} = F_{Sj} / k_{Sj} \tag{5}$$

In this mechanical assembly, force $F_{Sj}$ remains constant regardless of the stiffness of the sensor element $k_{Sj}$, while extension $\delta_{Sj}$ of the sensor element scales inversely with its stiffness $k_{Sj} k_{sj}$. This simple solution, where forces are constant and extensions change with sensor element stiffness, is a prime example of an assembly exhibiting force-based control.

In response to a bulk extension input ($\delta_0$), more complex behaviors are observed. We must first determine the total force $F$ across the assembly following Hooke's Law for springs in series:

$$F = \delta_0 \left( \frac{1}{k_{Sj} N_{Sj}} + \frac{1}{k_L N_L} \right)^{-1} \tag{6}$$

where the effective spring constant for the whole assembly is given by:

$$\left( \frac{1}{k_{Sj} N_{Sj}} + \frac{1}{k_L N_L} \right)^{-1} \tag{7}$$

Calculating $F_{Sj}$ and $\delta_{Sj}$ as before (*Equations 4 and 5*), it becomes apparent that both $F_{Sj}$ and $\delta_{Sj}$ will change depending on the relative abundance and stiffness of the sensor and linker elements.

By performing this operation over a variety of relative abundance and stiffness of the sensor and linker elements, we begin to understand what kinds of structural arrangements could give rise to force-controlled versus extension-controlled loading of FA components (*Figure 4—figure supplement 9D*). Specifically, the assemblies most biased toward extension-controlled behaviors involve linker elements that are much stiffer and/or in greater molecular abundance as compared to sensor elements. However, in the opposite scenario, force-controlled loading could be observed even following a bulk extension input. The sensitivity of the control paradigm to sensor and linker element stiffness and abundance indicates that

force-controlled and extension-controlled scenarios are potentially mutable and highly dependent upon protein, cellular, and mechanical contexts. Indeed, in our multiple control paradigm experiments, the *Control Metric* was not constant, although it was consistently in the positive extension-controlled regime where the variance in forces was much greater than that of extensions. For MEFs adhered to fibronectin-coated glass substrates (*Figure 4*), vinculin exhibits strongly extension-controlled behavior (*Figure 4—figure supplement 9D*, *Control Metric* = 3.2). Pharmacological inhibition of ROCK-mediated myosin II activity (Y-27632 treatment) caused only a slight decrease in the strength of extension-control (*Figure 4—figure supplement 9D*, *Control Metric* = 3.1). Preventing vinculin-talin interactions via A50I mutation and plating cells on softer substrates had more noticeable impacts on control paradigm (*Figure 4—figure supplement 9D*, *Control Metric* = 2.6 and 1.7, respectively). Despite these small variations, the overall robustness of vinculin extension-control suggests a fundamental regulatory mechanism by which molecular motors or other cytoskeletal machineries produce discrete displacements to load vinculin.

The model also predicts the relationship between sensor element stiffness $k_{Sj}$ and the force borne by the sensor element $F_{Sj}$. Importantly, for various values of relative abundance or stiffness of the sensor and linker elements, the two control regimes predict distinct relationships between sensor element stiffness $k_{Sj}$ and force $F_{Sj}$. To illustrate this, we depict the predicted stiffness-force relationships for the *Control Metric* values observed in our four control paradigm experiments (*Figure 4—figure supplement 9D*, dashed contour lines). In all cases, the extension-controlled regime is associated with a positive, monotonically increasing relationship between sensor element stiffness and force, although this relationship becomes more nonlinear in the case of VinTS-expressing MEFs on 10 kPa substrates where the *Control Metric* is reduced (*Figure 4—figure supplement 9E*). In contrast, model predictions and experimental measurements of extension-control indicate no relationship between sensor element stiffness and extensions, as extensions remain constant (*Figure 4—figure supplement 9F*). Note, the magnitudes of the forces and extensions must be considered qualitatively, as they are affected by the magnitude of the bulk extension input ($\delta_0$). Together, these model predictions indicate that the extension-controlled loading of individual FA components is most likely to be observed in response to a bulk extension input, will be observed for soft elements in relatively low abundance, and will result in a linear relationship between sensor element stiffness and load bearing capacity, as we observed experimentally for vinculin in a variety of mechanical and cellular contexts. Determining whether other load-bearing proteins are subject to extension-control will be critically important to understanding mechanotransduction mechanisms in cells.

