## [Decision Letter]

Thank you for submitting your article "Tunable molecular tension sensors reveal extension-based control of vinculin loading" for consideration by *eLife*. Apologies for the delay in getting back to you with this decision letter. Your article has been favorably evaluated by Anna Akhmanova (Senior Editor) and three reviewers, one of whom is a member of our Board of Reviewing Editors. The following individual involved in review of your submission has agreed to reveal his identity: Khalid Salaita (Reviewer #3).

The reviewers have discussed the reviews with one another and the Reviewing Editor has drafted this decision to help you prepare a revised submission.

Summary:

This manuscript describes the design of molecular tension sensor modules that, in combination with FRET, provide a much more controlled and calibrated manner to access intracellular forces and tensions. Particularly, the authors used unstructured synthetic domains as tension sensors and modeled their extension behavior as random coils to predict extension versus force. They use three of these sensors to measure and characterize vinculin loading. Based on these experiments, they design a large number of sensor elements that will find broad applicability in mechanobiology. All three reviewers commented on the high quality of the work and the significant methodological advance that the work represents.

Essential revisions:

The main biological conceptual advance, the finding that the activation of vinculin mechanotransduction signaling is governed by protein extension rather than force, is of significant interest but not convincingly supported by data. Performing every experiment described in the points below is outside the scope of the current study. However, to strengthen the central mechanistic claim in the work, additional experimental data are needed and some of the suggestions immediately below will need to be implemented.

- While cells do induce similar strains within environments of different stiffness within certain stiffness regimes, it has been well described that the same cell type may exhibit sustained traction or sustained strain within distinct environmental stiffness regimes. For the vinculin experiments described here, the environmental force input is fibronectin-coated glass. The supraphysiologic stiffness of glass certainly will simulate the maximal extreme of "linker abundance and stiffness" in the authors FA structural model and therefore predictively result in an extension-based vinculin activation. To further strengthen the model and the biological conclusions on vinculin mechanotransduction, the authors should conduct additional vinculin loading experiments with their trio of sensors on a range of substrates of physiologic stiffness to test whether vinculin is exclusively regulated by an extension-based paradigm, or if its activation can be also force-based.

- It is unclear how the authors' conclusions on the extension-based loading mechanism relate to native vinculin and the other mechanosensing proteins within the focal adhesion. If the force is reduced to maintain extension, then the other mechanosensitive proteins would not be adequately activated. Also, the innate auto-inhibited vinculin would not be fully activated if the force level is reduced (considering the work published by Garcia et al., 2013 "How vinculin regulates force transmission"). Finally, if this is true, then it seems likely that the genetically encoded tension sensors likely perturb the focal adhesion function and also mechanosensing as a whole since the differing linker lengths will perturb extension sensing. Perhaps there is another way to interpret this conclusion or to better describe the meaning of the results?

The "FA force slope" (Figure 3) should be applied across the different linker lengths to probe whether the force gradient or the gradient of extension is similar across the different probes. Another interesting metric would be to analyze time lapse videos of the cells with differing linker lengths to determine whether there are differences in the FRET (force) dynamics across the different linkers. It seems reasonable to suspect that both of these pieces of data are already available and it simply requires further analysis. The last suggestion is not absolutely essential but it would be very interesting to test whether the total cell traction forces are different as a function of the vinculin linker length. This is essentially an internal validation to test whether linker length influences traction forces and cell traction output.

---

## [Author Response]

Essential revisions:The main biological conceptual advance, the finding that the activation of vinculin mechanotransduction signaling is governed by protein extension rather than force, is of significant interest but not convincingly supported by data. Performing every experiment described in the points below is outside the scope of the current study. However, to strengthen the central mechanistic claim in the work, additional experimental data are needed and some of the suggestions immediately below will need to be implemented.- While cells do induce similar strains within environments of different stiffness within certain stiffness regimes, it has been well described that the same cell type may exhibit sustained traction or sustained strain within distinct environmental stiffness regimes. For the vinculin experiments described here, the environmental force input is firbronectin-coated glass. The supraphysiologic stiffness of glass certainly will simulate the maximal extreme of "linker abundance and stiffness" in the authors FA structural model and therefore predictively result in an extension-based vinculin activation. To further strengthen the model and the biological conclusions on vinculin mechanotransduction, the authors should conduct additional vinculin loading experiments with their trio of sensors on a range of substrates of physiologic stiffness to test whether vinculin is exclusively regulated by an extension-based paradigm, or if its activation can be also force-based.

We agree that further investigation into the biological limits of extension-based vinculin loading is important. To test whether vinculin is exclusively regulated by an extension-based paradigm, we conducted three additional experiments. Specifically, we examined if the observation of extension-control was maintained when 1) ROCK-mediated cell contractility was reduced by treating cells with Y-27632, 2) interactions between vinculin and talin (thought to be critical to loading of vinculin) were prevented by utilizing vinculin tension sensors harboring the A50I point mutation, or 3) cells were plated on a deformable substrate with a physiologically relevant stiffness (10 kPa).

Inhibition of ROCK-dependent cytoskeletal contractility caused the expected changes in FA shape, cell morphology and molecular loads across vinculin, as has been observed in our previous work (Rothenberg et al., 2018). Despite these morphological and mechanical changes, a control paradigm study utilizing the three distinct sensors showed that extensions across vinculin (not forces) were maintained (Figure 4—figure supplement 3, 4).

Next, we investigated the requirement for talin binding in extension sensing. To do so, we utilized VinTS A50I, which contains a point mutation that prevents talin binding. Cells expressing these sensors exhibited the expect changes in FA size and number, as well as a loss of the spatial gradients of loading in FAs (Rothenberg et al., 2018). Again, the control paradigm study revealed that extension-control of vinculin was maintained (Figure 4—figure supplement 5, 6).

Lastly, we plated cells on softer, 10kPa polyacrylamide gels coated with fibronectin. Interestingly, we did not detect a difference in the magnitude of vinculin loading, consistent with previous work on continuous PDMS substrates of comparable stiffness (Kumar et al., 2017). However, as this previous work only used a single tension sensor the existence of extension-control or force-control for vinculin could not be determined. Here, using three mechanically distinct sensors, we observed extension-control is maintained on substrates with physiologically-relevant stiffness (Figure 4—figure supplement 7, 8).

Together, these results strongly suggest that vinculin loading is exclusively mediated by an extension-based control mechanism in all of the conditions we probed. However, the strength of the extension-control, quantified by control metric, is variable (displayed as contour lines in Figure 4—figure supplement 9D). While all values indicate extension-control, this variation does suggest the possibility that there may be other conditions, or other proteins, that exhibit force-control. Overall, these data support the idea that vinculin is typically subject to extension-control, but there may be other specific conditions, or other proteins, that experience force-control.

Action taken: We have included the results of our VinTS + Y-27632 treatment, VinTS-A50I data, as well as the cells-on-gels experiments as supplemental figures to Figure 4. These are referenced in the Results section of the revised manuscript (subsection “Vinculin loading is subject to an extension-based control mechanism”, second paragraph) and the robustness of extension-based control of vinculin loading is discussed in the Discussion section (fifth paragraph).

- It is unclear how the authors' conclusions on the extension-based loading mechanism relate to native vinculin and the other mechanosensing proteins within the focal adhesion. If the force is reduced to maintain extension, then the other mechanosensitive proteins would not be adequately activated. Also, the innate auto-inhibited vinculin would not be fully activated if the force level is reduced (considering the work published by Garcia et al., 2013 "How vinculin regulates force transmission"). Finally, if this is true, then it seems likely that the genetically encoded tension sensors likely perturb the focal adhesion function and also mechanosensing as a whole since the differing linker lengths will perturb extension sensing. Perhaps there is another way to interpret this conclusion or to better describe the meaning of the results?

We agree that the ramifications of an extension-based loading mechanism for endogenous vinculin could be discussed with greater clarity and detail. Our results are consistent with a model where, once vinculin is activated, it is pulled until a given extension is achieved, not a set force. Note that “extension” refers to the change in length of the polypeptide spring, not the total length of the spring. For instance, the rest lengths (end-to-end distance at F = 0 pN) of the various springs in the tension sensors used here are 3.04, 3.62, and 4.12 nm. Despite these differences, all sensors are subject to extension of ~3 nm in control cells (although slightly different extensions are observed in other experiments). To investigate the mechanisms underlying extension-control, we performed simulations of simple mechanical models of FAs connected to the cytoskeleton. These analyses suggest that cytoskeleton is applying discrete extensions to enable this type of mechanism.

In regards to vinculin’s biochemical activation state, there is limited evidence to suggest that force application leads to the direct activation of vinculin. The available data, most of which is based on work with VinTS, shows that vinculin loading and activation are separable (Grashoff et al., 2010; Rothenberg et al., 2018). Furthermore, to our understanding, the work by Garcia and colleagues does not show that force is required for vinculin activation. It argues that force facilitates “the stabilization of the activated state”. This picture is consistent with our results, except we would add that the extension of vinculin, not the force it supports, is the key mechanical variable for maintaining it in the activated state.

Also, we observe no evidence that the various sensors perturb FA function. Alterations in vinculin function are known to cause distinctive changes is FA characteristics. Since we did not detect differences in FA size, intensity, or number between the various constructs (Figure 3—figure supplement 2) and each construct responded similarly to Y-27632 treatment, A50I mutation, and 10 kPa gels (Figure 4—figure supplements 4, 6, 8), we conclude that the tension sensors are exhibiting indistinguishable biological functions. Furthermore, we note that despite the different linker “rest” lengths, all tension sensors are extended by the same amount (see Figure 4—figure supplement 1D). The existence of this extension sensing indicates that FA function is not perturbed, since this seems to be the hallmark of vinculin’s mechanical regulation. In total, the simplest model that encompasses all of this data is that all sensors are biologically functional and subject to extension-control, and that sensors with different lengths do not perturb extension sensing.

An important question raised by the reviewer is if other proteins are subject to extension-control or force-control. Here we show that vinculin is clearly subject to robust extension-control. However, this may not be the case for all proteins in the FA. Also, recent work by Grashoff and colleagues has shown that the forces experienced by talin vary across the protein itself (Ringer et al., 2017). This suggests that the mechanical environment within the FA is significantly more complicated than currently appreciated. The approaches developed here for creating sensors with diverse mechanical sensitivities as well as those for determining if specific proteins are subject to extension-control or force-control are generalizable to any tension sensor and will be critically important in elucidating and understanding this complexity.

Action taken: To improve clarity and provide context for these findings, we have re-written and expanded the entire Discussion section of the revised manuscript. Specifically, we strived to re-focus the discussion of vinculin extension-control to talk about the role of the cytoskeleton and how our findings might relate to or impact other FA proteins (Discussion, fifth paragraph). We have also added our interpretation of extension-control in terms of the function of native vinculin, providing clarification as to the role of mechanical force in vinculin activation and further force-dependent processes (Discussion, sixth paragraph). We end with a brief discussion of extension-based versus force-based control as it relates to interpretation of tension sensor data (Discussion, seventh paragraph). We have also clarified our definitions of sensor “force” and “extension” in the Results section (subsection “Vinculin loading is subject to an extension-based control mechanism”, first paragraph) as well as in the figure caption and diagrams in Figure 4—figure supplement 1.

The "FA force slope" (Figure 3) should be applied across the different linker lengths to probe whether the force gradient or the gradient of extension is similar across the different probes. Another interesting metric would be to analyze time lapse videos of the cells with differing linker lengths to determine whether there are differences in the FRET (force) dynamics across the different linkers. It seems reasonable to suspect that both of these pieces of data are already available and it simply requires further analysis. The last suggestion is not absolutely essential but it would be very interesting to test whether the total cell traction forces are different as a function of the vinculin linker length. This is essentially an internal validation to test whether linker length influences traction forces and cell traction output.

We thank the reviewer for these helpful suggestions and have performed the requested analysis for which data was available (FA force slope). We agree with the reviewer that time lapse movies would be interesting, but they are not essential to demonstrate the claims in the paper. We also agree that a comparison of traction force microscopy and tension sensors would be interesting. However, traction force microscopy measures the entire stress generated by a cell while tension sensors measure the forces experienced by a single protein. Given that there are multiple load-bearing proteins within the FA (e.g. talin, ILK, filamin), comparisons between these techniques are not straightforward and cannot be used as internal validations. Since these experiments are not central to the claims of the paper, we consider them beyond the scope of this manuscript. We do thank the reviewer for the insightful comments and will try to investigate these possibilities in future work.

Action taken: Line scans of acceptor intensity, FRET, forces, and extensions reported by the three versions of VinTS were compared. Like global histogram analysis, line scans of highly loaded peripheral FAs showed different forces but similar extensions borne by each of these three sensors, further supporting the extension-control finding (Figure 4—figure supplement 2, referred to in the second paragraph of the subsection “Vinculin loading is subject to an extension-based control mechanism”).